# Learning Neuron Dynamics within Deep Spiking Neural Networks

## Abstract

Spiking Neural Networks (SNNs) offer a promising energy-efficient alternative to Artificial Neural Networks (ANNs) by utilizing sparse and asynchronous processing through discrete spike-based computation. However, the performance of deep SNNs remains limited by their reliance on simple neuron models, such as the Leaky Integrate-and-Fire (LIF) model, which cannot capture rich temporal dynamics. While more expressive neuron models exist, they require careful manual tuning of hyperparameters and are difficult to scale effectively. This difficulty is evident in the lack of successful implementations of complex neuron models in high-performance deep SNNs. In this work, we address this limitation by introducing Learnable Neuron Models (LNMs). LNMs are a general, parametric formulation for non-linear integrate-and-fire dynamics that learn neuron dynamics during training. By learning neuron dynamics directly from data, LNMs enhance the performance of deep SNNs. We instantiate LNMs using low-degree polynomial parameterizations, enabling efficient and stable training. We demonstrate state-of-the-art performance in a variety of datasets, including CIFAR-10, CIFAR-100, ImageNet, and CIFAR-10 DVS. LNMs offer a promising path toward more scalable and high-performing spiking architectures.

## 1 Introduction

Artificial Neural Networks (ANNs) have seen widespread adoption due to their remarkable success in areas such as computer vision (Chai et al., 2021) and natural language processing (Khan et al., 2023), among others. However, this success has come with a substantial increase in energy consumption (Yamazaki et al., 2022). In response, Spiking Neural Networks (SNNs) have emerged as a promising energy-efficient alternative. Unlike ANNs, which rely on continuous, synchronous computations, SNNs operate asynchronously using discrete, sparse events known as spikes. This spiking behavior is typically driven by biologically inspired neuron models (Gerstner et al., 2014). The event-driven and spiking nature of SNNs enables them to mimic the brain's sparse connectivity and energy-efficient behavior.

When mapped onto hardware designed to exploit these characteristics, SNNs can achieve significantly lower energy consumption than traditional ANNs (Rathi & Roy, 2023). Such hardware, known as neuromorphic hardware, is designed to support the sparse and event-driven computation style of SNNs. Notable examples include IBM's TrueNorth chip (Akopyan et al., 2015), used for always-on speech recognition in edge devices (Tsai et al., 2017), and Intel's Loihi 2 (Intel, 2021), which enables ultra-low-power image classification tasks (Lenz et al., 2023).

In deep spiking neural networks, the temporal evolution of a neuron's membrane potential is typically governed by a fixed neuron model. The Leaky Integrate-and-Fire (LIF) neuron model, defined by a linear leakage of membrane potential over time, has become the standard due to its simplicity and stability (Gerstner et al., 2014; Wu et al., 2018). However, the expressiveness of the LIF model is limited, as it cannot capture the diverse range of nonlinear temporal behaviors observed in biological neurons. More sophisticated neuron models—such as the Quadratic Integrate-and-Fire, Exponential Integrate-and-Fire, and Adaptive Exponential models—offer richer dynamics and enhanced modeling capacity. Despite their theoretical advantages, these models introduce added complexity and additional hyperparameters (e.g., thresholds, adaptation currents, non-linear scaling factors) that are difficult to tune and often require domain-specific knowledge. As a result, their use in deep SNN

architectures remains limited, with only the LIF model demonstrating consistent success in scalable, high-performance models.

The reliance on simple neuron models, such as LIF, can limit a network's performance, as they fail to capture the rich temporal structure required for complex tasks. Motivated by this limitation, we advocate for a shift from simple neuron dynamics and propose Learnable Neuron Models (LNM), in which a neuron's dynamics are parameterized and optimized alongside the rest of the network. By making a neuron's dynamics task-adaptive, learnable neuron models enhance the performance of SNNs without the difficulties associated with integrating sophisticated neuron models. To summarize, the main contributions of our work are as follows:

- We derive a general parametric formulation that encompasses the family of nonlinear integrate-and-fire neuron models, allowing arbitrary differentiable functions to be integrated into the membrane dynamics and learned during training. We call our formulation **Learnable Neuron Models (LNM)**.

- We realize **Learnable Neuron Models** through low-degree polynomial parameterizations, enabling neuron dynamics to be learned directly from data. By leveraging Horner's method for polynomial evaluation, our approach introduces negligible energy overhead compared to LIF.

- Our method achieves state-of-the-art accuracy compared to LIF-based convolutional deep SNNs on both static and neuromorphic datasets, with results of $97.01\%$ on CIFAR-10, $80.70\%$ on CIFAR-100, and $81.39\%$ on CIFAR10-DVS using ResNet-19 and $70.91\%$ on ImageNet using ResNet-34 surpassing the previous best results by $0.54\%$, $0.50\%$, $1.55\%$, and $0.17\%$, respectively.

## 2 RELATED WORK

### 2.1 DEEP LEARNING WITH SPIKING NEURAL NETWORKS

Training spiking neural networks directly with backpropagation is challenging. This is mainly due to the non-differentiability of spiking behavior. In recent years, several techniques have been developed to overcome this issue. One of the most popular techniques is called surrogate gradients. Instead of calculating the exact gradient of spiking behavior in SNNs, surrogate gradients provide approximations that can be used for backpropagation. Training performance largely depends on the quality of this approximation. Additionally, to make training SNNs with surrogate gradients stable and improve their performance, various techniques have been developed. These include developing new loss functions (Deng et al., 2022; Guo et al., 2022; Shen et al., 2024), normalization techniques (Zheng et al., 2021; Duan et al., 2022; Guo et al., 2023), and learning optimal surrogate gradients (Li et al., 2021; Lian et al., 2023; Deng et al., 2023).

We want to note that, due to the lack of support for training on neuromorphic datasets when using ANN-to-SNN conversion techniques (Cao et al., 2015), we restrict all comparisons in this work to direct training techniques with surrogate gradients.

### 2.2 NEURON MODELS AND PARAMETER LEARNING

Several works that employ direct training techniques, such as Fang et al. (2021); Yao et al. (2022); Rathi & Roy (2023); Lian et al. (2023; 2024), aim to modify the LIF neuron to improve model performance. For example, Fang et al. (2021) proposes a trainable decay factor of the LIF neurons that is independently learned for each layer. Rathi & Roy (2023) builds upon this idea by utilizing a learnable threshold in addition to the decay factor. Yao et al. (2022) proposed modifications to the LIF neuron, which make it act similar to long-short term memory by utilizing a gating mechanism that can choose the optimal biological features during training. Similarly, Zhang et al. (2024) and Feng et al. (2025) introduce two-compartment LIF neurons for long-term sequence modeling. Huang et al. (2025) propose a extra paths to facilitate better temporal backpropagation. Lian et al. (2023) proposes dynamically adjusting the surrogate gradient window during training when using a learnable decay factor to minimize gradient mismatch. Lian et al. (2024) proposes a temporal-wise attention mechanism, allowing neurons to establish connections with past temporal data selectively.

Guo et al. (2024b) aims to reduce information loss of binary spiking behavior by utilizing ternary spikes. Additionally, the authors parameterize their ternary spike with a learnable constant, allowing neurons to fire real-valued spikes.

In this work, we propose enhancing SNN performance by learning the dynamics of neurons within each network layer. To the best of our knowledge, this area of SNNs has not yet been explored.

## 3 BACKGROUND

### 3.1 SPIKING NEURAL NETWORKS

While ANNs use continuous-valued data to transmit information, SNNs use discrete events, known as spikes. The choice of neuron model governs the spiking dynamics of an SNN. In modern deep SNN research, non-linear integrate-and-fire neuron models are typically preferred for their blend of computational efficiency and biologically plausible dynamics (Gerstner et al., 2014). The general form for all non-linear integrate-and-fire neuron models is

$$\tau \frac{du}{dt} = f(u) + RI. \tag{1}$$

where $\tau$ is a membrane time constant, $u$ is the membrane potential, the function $f(u)$ governs the intrinsic dynamics of $u$, $R$ is a linear resistor, and $I$ is pre-synaptic input. By changing the function $f(u)$, we obtain different neuron models within the non-linear integrate-and-fire family. For example, if we define $f(u) = u_r - u$, for some resting potential $u_r$, we obtain the LIF neuron model.

To utilize non-linear integrate-and-fire neuron models in deep learning scenarios, discretization is required (Duan et al., 2022). The most commonly used discretization technique is Euler's method (Wu et al., 2018). Applying Euler's method to Equation 1, we obtain

$$u(t + 1) = u(t) + \frac{\Delta t}{\tau}(f(u(t)) + RI(t)), \tag{2}$$

where $t$ is the timestep and $\Delta t$ is the discretization step size. To simplify Equation 2, we can fold the term $\frac{\Delta t}{\tau}R$ into the pre-synaptic weights of $I$ to obtain

$$u(t + 1) = u(t) + \frac{\Delta t}{\tau}f(u(t)) + I(t). \tag{3}$$

When $u(t + 1)$ exceeds a threshold $u_{th}$, a spike is produced and the voltage is reset as follows,

$$u(t + 1) = [u(t) + \frac{\Delta t}{\tau}f(u(t))](1 - o(t)) + I(t) \tag{4}$$

$$o(t + 1) = \Theta(u(t + 1) - u_{th}), \tag{5}$$

where $\Theta$ is the Heaviside function given by $\Theta(x) = 0$ if $x < 0$, else $\Theta(x) = 1$. Note, the term $(1 - o(t))$ in Equation 4 corresponds to a hard reset. Equation 4 and Equation 5 enable forward and backward propagation in spatial and temporal domains (Zheng et al., 2021).

### 3.1.1 TRAINING SPIKING NEURAL NETWORKS

The Spatial-Temporal Back Propagation (STBP) algorithm is commonly used to train SNNs (Wu et al., 2018). First, an SNN processes temporal data for $T$ timesteps. The SNN's output is decoded by accumulating the synaptic voltage of the last layer as follows

$$\hat{y} = \frac{1}{T} \sum_{t=1}^{T} W o(t). \tag{6}$$

In the above equation, $W$ is a weight matrix, $o(t)$ is the spiking activity at timestep $t$, and $\hat{y} \in \mathbb{R}^m$ is the model output given $m$ classes. Using our output vector $\hat{y} = (\hat{y}_1, \hat{y}_2, \ldots, \hat{y}_m)$ and a label vector

$y = (y_1, y_2, \ldots, y_m)$, we compute the cross entropy loss, $L$, between $\hat{y}$ and $y$. Then, using the STBP algorithm, we train the network using the chain rule to update synaptic weights by

$$\frac{\partial L}{\partial W_n} = \sum_{t=1}^{T} \left[ \frac{\partial L}{\partial o_n(t+1)} \frac{\partial o_n(t+1)}{\partial u_n(t+1)} + \frac{\partial L}{\partial u_n(t+1)} \frac{\partial u_n(t+1)}{\partial u_n(t)} \right] \frac{\partial u_n(t+1)}{\partial W_n}, \tag{7}$$

where $n$ is the layer of the network, $u(t)$ is a neuron's membrane potential, and $o(t)$ is a neuron's spike output (Guo et al., 2023).

To overcome the undefined derivative, $\frac{\partial o_n(t+1)}{\partial u_n(t+1)}$, Wu et al. (2018) proposed using the derivative of an approximation to the Heaviside function with useful gradient information. This technique is known as a surrogate gradient. A common choice of surrogate gradient is the rectangle function (Zheng et al., 2021; Deng et al., 2022; Lian et al., 2023), which is defined by

$$\frac{\partial o(t+1)}{\partial u(t+1)} \approx \frac{1}{\alpha} \text{sign} \left( |u(t+1) - u_{th}| < \frac{\alpha}{2} \right), \tag{8}$$

where $\alpha$ determines the width and area of the surrogate gradient and typically remains constant throughout training. The choice of $\alpha$ has a significant impact on the learning process of SNNs, with improper choices leading to gradient mismatches and approximation errors.

## 4 METHODOLOGY

### 4.1 LIMITATIONS OF FIXED NEURON MODELS

In deep spiking neural networks, a single function $f(u)$ is typically chosen to govern the membrane dynamics of all neurons in the network. The LIF neuron model is the most widely adopted due to its computational efficiency and simplicity. Although more expressive models—such as the quadratic integrate-and-fire, exponential integrate-and-fire, and adaptive exponential integrate-and-fire neurons—are available, they require careful manual tuning of hyperparameters, which often vary across tasks and datasets. In practice, identifying stable and effective hyperparameters for training deep SNNs with such neuron models remains a significant and unsolved challenge. This problem is evident as, to date, only the LIF model has shown success in high-performance deep SNNs.

### 4.2 LEARNABLE NEURON MODELS

To address this issue, we propose a parameterization of a neuron's dynamics with a set of learnable weights $\theta \in \mathbb{R}^N$, yielding a function $f_\theta(u)$. Specifically, each layer of an SNN can learn a unique set of weights $\theta$, thereby increasing the network's expressiveness and enabling it to learn the optimal neuron models for a given task.

Incorporating $f_\theta(u)$ into Equation 2 provides us with

$$u(t+1) = u(t) + \frac{\Delta t}{\tau} f_\theta(u(t)) + I(t). \tag{9}$$

We can then fold the constant factor $\frac{\Delta t}{\tau}$ into $\theta$, resulting in a simplified update rule,

$$u(t+1) = u(t) + f_\theta(u(t)) + I(t). \tag{10}$$

#### 4.2.1 CHOICE OF PARAMETERIZATION

A key choice in our methodology lies in selecting a suitable parameterization for $f_\theta$, which defines the subthreshold dynamics of the neuron. This choice is guided by three primary constraints: computational efficiency, expressiveness, and differentiability. We evaluate four function families under these criteria: multi-layer perceptrons (MLPs) (Minsky & Papert, 2017), splines (Schumaker, 2007), Chebyshev polynomials (Hale, 2015), and polynomials.

*MLPs* are arguably the most expressive choice of function. Their universal approximation capability enables them to represent highly nonlinear and task-specific dynamics with a relatively small number

of parameters. However, MLPs are not suitable for neuromorphic hardware due to their continuous-valued, synchronous data processing.

*Splines* provide a compelling middle ground between MLPs and polynomials. Cubic splines, in particular, can provide piecewise-smooth approximations with local control, which can be advantageous for modeling neuron dynamics that exhibit distinct behavioral regimes. However, in our testing, splines introduced the greatest computational overhead due to the difficulty of efficiently vectorizing their operations.

*Chebyshev Polynomials* offer a promising trade-off among the three constraints; however, low-degree variants, necessary for computational efficiency, tend to exhibit undesirable sinusoidal behavior due to their orthogonal basis. This, in turn, constrains the dynamics of a neuron. Additionally, Chebyshev polynomials require their input to be scaled. This can be a costly operation and not one typically suited to neuromorphic processors (Intel, 2021).

*Polynomials* contrast with Chebyshev polynomials in that low-degree polynomials don't necessarily exhibit a specific behavior. While polynomials may require high degrees to obtain the same expressivity as the other function families, they are the most computationally efficient. Additionally, polynomials align with the dynamics of existing neuron models, such as the LIF and QIF. Consequently, we adopt polynomials as the most practical parameterization, defining $f_\theta$ as

$$f_\theta(u(t)) = \sum_{i=0}^{N} \theta_i u(t)^i. \tag{11}$$

Our use of polynomial functions introduces two key challenges. First, while low-degree polynomials are more favorable for computational efficiency, they limit the expressiveness of $f_\theta$. Second, polynomial functions exhibit unbounded growth outside a limited input range, which can potentially lead to instability during training or inference. To mitigate both issues, we clip the input to $f_\theta$ to the range $[-1, 1]$. Clipping ensures that values below $-1$ and above $1$ are set to $-1$ and $1$, respectively. This bounding improves the numerical stability of polynomials. Additionally, working within this tighter interval enables better usage of low-degree polynomials, increasing their expressiveness.

### 4.2.2 TRAINING

To train the learnable weights of LNM using STBP, we use our updated neuron logic in Equation 10 and define the derivative of a weight $\theta_k$ with respect to a loss function $L$ as

$$\frac{\partial L}{\partial \theta_k} = \sum_{t=1}^{T} \left[ \frac{\partial L}{\partial o_n(t+1)} \frac{\partial o_n(t+1)}{\partial u_n(t+1)} + \frac{\partial L}{\partial u_n(t+1)} \frac{\partial u_n(t+1)}{\partial f_\theta(u_n(t))} \right] \frac{\partial f_\theta(u_n(t))}{\partial \theta_k} \tag{12}$$

In the equation above, $n$ is the network layer, and $u(t)$ and $o(t)$ are the neuron's membrane potential and spike output at timestep $t$, respectively. Additionally, we need to redefine the gradient of synaptic weight $W_n$ as

$$\frac{\partial L}{\partial W_n} = \sum_{t=1}^{T} \left[ \frac{\partial L}{\partial o_n(t+1)} \frac{\partial o_n(t+1)}{\partial u_n(t+1)} + \frac{\partial L}{\partial u_n(t+1)} \left( 1 + \frac{\partial f_\theta(u_n(t))}{\partial u_n(t)} \right) \right] \frac{\partial u_n(t+1)}{\partial W_n}. \tag{13}$$

Using Equation 12 and Equation 13, STBP can be implemented with our proposed learnable neuron model as long as $f_\theta$ is differentiable.

### 4.2.3 EFFICIENT POLYNOMIAL EVALUATION

To efficiently evaluate the polynomial $f_\theta$, we utilize Horner's method (Sutin, 2008). That is, we iteratively factor $u(t)$ terms out of Equation 11. For example, we can evaluate a third-degree polynomial using Horner's method as follows:

$$f_\theta(x) = \theta_0 + \theta_1 u(t) + \theta_2 u(t)^2 + \theta_3 u(t)^3 \tag{14}$$
$$= \theta_0 + u(t)(\theta_1 + u(t)(\theta_2 + \theta_3 u(t))) \tag{15}$$

Horner's method enables evaluation of an $N$-degree polynomial in $\mathcal{O}(N)$ time, compared to the $\mathcal{O}(N^2)$ time required by naïve evaluation.

If $f_\theta$ is at least a first-degree polynomial, then we can further simplify Equation 10. That is, for an N-degree polynomial, with $N \geq 1$, observe

$$
\begin{aligned}
u(t+1) &= u(t) + f_\theta(u(t)) + I(t) \\
&= u(t) + \theta_0 + \theta_1 u(t) + \cdots + \theta_N u(t)^N + I(t) \\
&= \theta_0 + (1+\theta_1)u(t) + \cdots + \theta_N u(t)^N + I(t) \\
&= f_\theta(u(t)) + I(t).
\end{aligned}
\tag{16}
$$

This simplification removes one addition otherwise required to evaluate $u(t+1)$.

Post-training, techniques such as least-squares polynomial approximation (Guo et al., 2020) can be used to find a lower-degree polynomial to approximate the learned polynomial and further reduce computational complexity. This can be applied in scenarios when an LNM has been overparameterized or when weight regularization techniques are applied.

## 5 EXPERIMENTS

We test our proposed learnable neuron model and compare it to recent state-of-the-art works. We utilize widely-adopted model architectures and datasets, such as ResNet-19 (Zheng et al., 2021) for CIFAR-10 and CIFAR-100 (Krizhevsky, 2009), ResNet-34 (Zheng et al., 2021) for ImageNet, and ResNet-19 (Zheng et al., 2021) and VGGSNN (Deng et al., 2022) for CIFAR-10 DVS (Krizhevsky, 2009). For all experiments, we compare the average top-1 validation accuracy for each model across three training runs. To promote stability during training, we initialize each LNM as the LIF neuron model and constrain $f_\theta(0) = 0$ for all choices of $\theta$. We train an LIF neuron model under the same conditions to provide a baseline for comparison. Lastly, we adopt a similar hyperparameter choice as used in the recent work of Shen et al. (2024). Our exact hyperparameters and dataset augmentations are detailed in Appendix A.

### 5.1 ABLATION STUDY

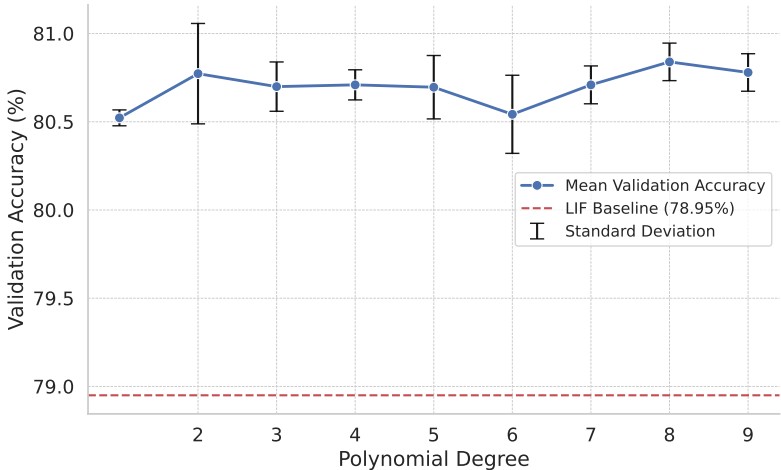

Figure 1: Ablation study on polynomial degree.

To determine the optimal polynomial degree, we conduct an ablation study using a ResNet-19 architecture trained on CIFAR-100 with four timesteps. For each polynomial degree, we perform three independent training runs and report the mean and standard deviation to assess stability and performance. Additionally, we include a baseline of the LIF neuron's performance under the same training conditions. As shown in Figure 1, our learnable neuron model does benefit from higher-degree polynomials. However, beyond third-degree polynomials, the mean validation accuracy starts to plateau,

Table 1: Summary and comparison of accuracy results on static datasets. The **Timesteps** column correlates to **CIFAR-10/CIFAR-100/ImageNet**. All **CIFAR-10/CIFAR-100** results utilize ResNet-19 while all **ImageNet** results utilize ResNet-34. Acronyms: Surrogate Gradient (SG), Learnable Neuron Model (LNM). Initial Membrane Potential (IMP).

| Work | Method | Timesteps | CIFAR-10 | CIFAR-100 | ImageNet |
|---|---|---|---|---|---|
| TET Deng et al. (2022) | Loss Function | 6/6/6 | 94.50% | 74.72% | 68.00% |
| IM-Loss Guo et al. (2022) | Loss Function + SG | 2/2/6 | 93.85% | 70.18% | 67.43% |
| LocalZO + TET Mukhoty et al. (2023) | Direct Training | 2/2/× | 95.03% | 76.36% | × |
| Surrogate Module Deng et al. (2023) | Hybrid | 4/4/4 | 95.54% | 79.18% | 68.25% |
| MPBN Guo et al. (2023) | Membrane Normalization | 2/2/4 | 96.47% | 79.51% | 64.71% |
| Backpropagation Shortcuts Guo et al. (2024a) | Direct Training | 2/2/4 | 95.19% | 77.56% | 67.90% |
| IMP + LTS Shen et al. (2024) | IMP + Loss Function | ×/×/4 | × | × | 68.90% |
| PLIF Fang et al. (2021) | Neuron Model | 8/×/× | 93.50% | × | × |
| GLIF Yao et al. (2022) | Neuron Model | 4/4/4 | 94.85% | 77.05% | 67.52% |
| | | 2/2/× | 94.44% | 75.48% | × |
| LSG Lian et al. (2023) | Neuron Model + SG | 4/4/× | 95.17% | 76.85% | × |
| | | 2/2/× | 94.41% | 76.32% | × |
| IM-LIF Lian et al. (2024) | Neuron Model + Loss Function | 3/3/× | 95.29% | 77.21% | × |
| Ternary Spike Guo et al. (2024b) | Neuron Model | 2/2/4 | 95.80% | 80.20% | 70.74% |
| LIF (Our Implementation) | Baseline | 4/4/4 | $96.20 \pm 0.20\%$ | $78.95 \pm 0.02\%$ | 69.04% |
| | | 2/2/× | $96.16 \pm 0.01\%$ | $79.92 \pm 0.10\%$ | × |
| **LNM (Ours)** | **Neuron Model** | **4/4/4** | $\mathbf{97.01 \pm 0.04\%}$ | $\mathbf{80.70 \pm 0.14\%}$ | $\mathbf{70.91\% \pm 0.03\%}$ |
| | | **2/2/×** | $\mathbf{96.96 \pm 0.10\%}$ | $\mathbf{80.07 \pm 0.13\%}$ | × |

Table 2: Comparison between state-of-the-art techniques and our Learnable Neuron Model (LNM) on CIFAR-10 DVS.

| Dataset | Work | Method | Architecture | Timesteps | Accuracy |
|---|---|---|---|---|---|
| CIFAR-10 DVS | TET Deng et al. (2022) | Loss Function | VGGSNN | 10 | 77.40% |
| | IM-Loss Guo et al. (2022) | Loss Function + SG | ResNet-19 | 10 | 72.60% |
| | LocalZO + TET Mukhoty et al. (2023) | Direct Training | VGGSNN | 10 | 75.62% |
| | MPBN Guo et al. (2023) | Membrane Normalization | ResNet-19 | 10 | 74.40% |
| | GLIF Yao et al. (2022) | Neuron Model | ResNet-19 | 16 | 78.10% |
| | PSN Fang et al. (2023a) | Neuron Model | VGGSNN | 10 | 85.90% |
| | LSG Lian et al. (2023) | Neuron Model + SG | VGGSNN | 10 | 77.90% |
| | IM-LIF Lian et al. (2024) | Neuron Model + Loss Function | VGGSNN | 10 | 80.50% |
| | Ternary Spike Guo et al. (2024b) | Neuron Model | ResNet-19 | 10 | 79.84% |
| | LIF (Our Implementation) | Baseline | VGGSNN | 10 | $75.10 \pm 1.31\%$ |
| | | | ResNet-19 | 10 | $73.16 \pm 1.37\%$ |
| | **LNM (Ours)** | **Neuron Model** | **VGGSNN** | **10** | $\mathbf{82.95 \pm 0.69\%}$ |
| | | | **ResNet-19** | **10** | $\mathbf{81.39 \pm 0.25\%}$ |

and the standard deviation overlaps with higher-degree polynomials. Notably, while second-degree polynomials achieve slightly higher mean accuracy, they exhibit the largest variability, indicating the least stable performance. Consequently, we select third-degree polynomials as the optimal choice, offering a favorable trade-off between accuracy, stability, and computational efficiency. Our code base, included within the supplementary material, details the exact hardware and software system configurations used to run each experiment.

## 5.2 Comparison to Recent Works

First, we compare our method to the state-of-the-art (SOTA) on CIFAR-10. Looking at Table 1, the best previous result achieves $96.47\%$ accuracy. Our LNM approach surpasses this, reaching $96.96\%$ and $97.01\%$ with two and four timesteps, respectively. On CIFAR-100, our method achieves $80.07\%$ with two timesteps and $80.70\%$ with four timesteps. The prior best result reports $80.20\%$ with two timesteps (Guo et al., 2024b).

We next evaluate our LNM approach on the ImageNet dataset. Our technique achieves an accuracy of $70.91\%$, surpassing the previous state-of-the-art Ternary Spike method of Guo et al. (2024b) by $0.17\%$. Compared to the best-performing binary spike method by Shen et al. (2024), our approach achieves a $2.01\%$ higher accuracy. These results demonstrate the scalability of our method.

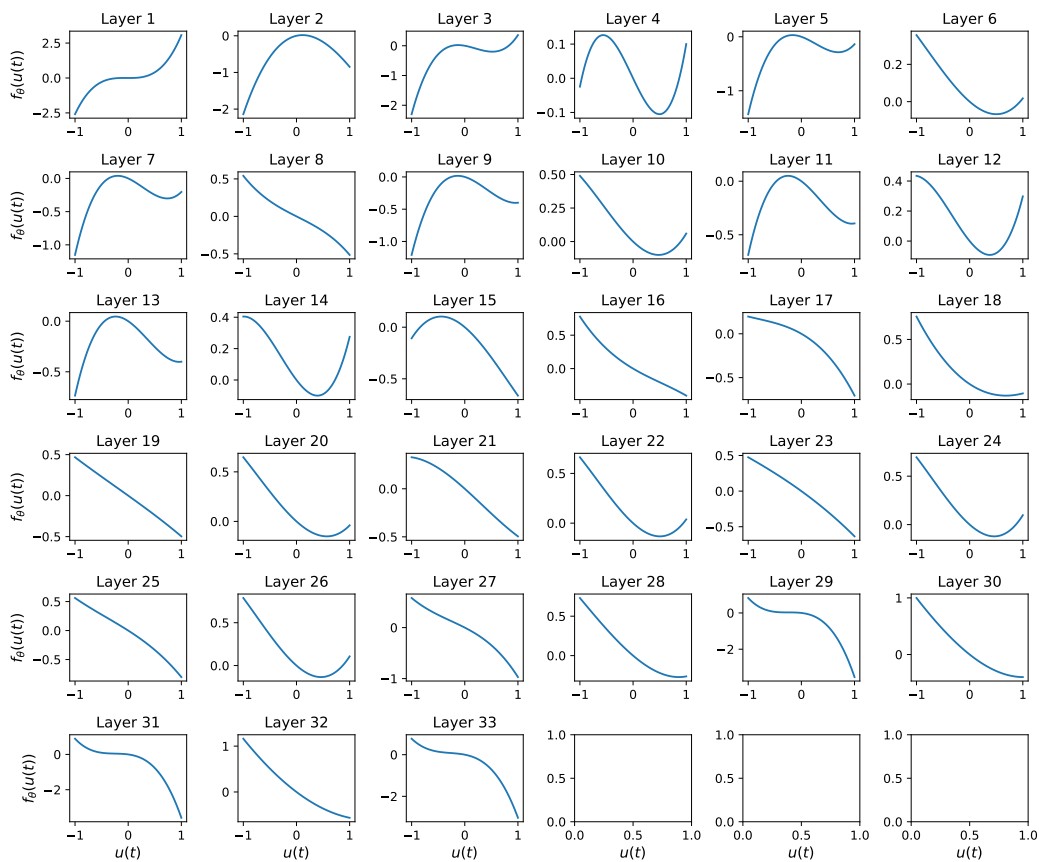

Figure 2: Learned neuron models of ResNet-34 trained on ImageNet with four timesteps.

Finally, we evaluate our method on CIFAR-10 DVS using VGGSNN and ResNet-19. In Table 2, we see VGGSNN trained with our method achieves 82.95% accuracy. While the work of Fang et al. (2023b) outperforms our LNM, we achieve a notable gain of 2.45% over Lian et al. (2024). Using ResNet-19, we improve on the prior state-of-the-art by 1.55%, achieving 81.39%. These results affirm that incorporating more expressive neuron dynamics can significantly enhance the performance of deep SNNs. Note that we do not compare against the work of Duan et al. (2022). Their method introduces Temporal Effective Batch Normalization, which cannot be folded into model weights after training.

## 5.3 LEARNED NEURON MODEL ANALYSIS

We examine the learned neuron models within a ResNet-34 architecture, as shown in Figure 2. This figure illustrates the distinct neuron models learned across each spiking layer of the network. Noting that we utilized a threshold of 0.5 for these experiments, only a small portion of layers (8, 10, 19, 21, 23, 25) learned neuron models that resemble LIF dynamics. Layers 6, 12, 14, 20, 22, 24, and 26 all learned dynamics similar to Quadratic Integrate-and-Fire (QIF) neurons (Gerstner et al., 2014). A few layers (2, 7, 9, 11, 13, and 15) learned dynamics that resemble the behavior of negative QIF neurons, a phenomenon not observed in biological neurons (Gerstner et al., 2014). Other layers (18, 28, 30, 32) interestingly learned negative Exponential Integrate-and-Fire (ExLIF) behavior (Gerstner et al., 2014). The remaining layers all learned dynamics that don't resemble any known neuron models. For example, layer 4 learned sinusoidal behavior, while layers 1, 3, 5, 29, 31, and 33 learned cubic behavior. Layers 16, 17, and 27 demonstrate logarithmic behavior. This diversity suggests that deep SNNs do benefit from a richer variety of neuron behaviors.

## 5.4 Energy Efficiency and Runtime

We calculate the average hardware energy cost per inference of LNMs. Note that the first layer of the network employs floating-point operations (FLOPS), while the remaining layers utilize synaptic operations (SOPS). We follow the technique used by Guo et al. (2024b) and calculate the energy cost of each layer $E_i$ as

$$E_i = T \cdot (fr \cdot E_{\mathrm{AC}} \cdot OP_{\mathrm{AC}} + E_{\mathrm{MAC}} \cdot OP_{\mathrm{MAC}}). \tag{17}$$

In the above equation, $T$ is the number of timesteps, $fr$ is the average spike-rate of neurons in layer $i$, $OP_{\mathrm{AC}}$ and $OP_{\mathrm{MAC}}$ are the number of accumulations and multiply-and-accumulate operations, respectively, with $E_{\mathrm{AC}}$ and $E_{\mathrm{MAC}}$ being their corresponding energy cost. We assume these operations take place with 32-bit floating point values on 45nm technology where $E_{\mathrm{MAC}} = 4.6pJ$ and $E_{\mathrm{AC}} = 0.9pJ$ (Lian et al., 2024). We compare the energy cost of both LNM and the LIF neuron in Table 3, assuming the same spike rate for both. It can be seen that our third-degree LNM only introduces energy consumption overheads between $2-5.5\%$ compared to LIF. Note that the LIF neuron requires a singular MAC operation per neuron update, while a degree-$N$ LNM requires $N$ MAC operations.

Lastly, we discuss the training and inference overheads observed in our experimental setup for degree-3 LNMs. Across all experiments, training incurred an average runtime overhead of approximately 42%, while inference exhibited a much smaller runtime overhead of around 4%. These results indicate that the additional learnable parameters non-trivially impact training time. However, since inference constitutes the primary workload of a model during deployment, the relatively modest increase in inference time suggests that the added complexity of LNMs remains practical for real-world applications.

It is worth noting that this inference overhead should be viewed as an upper bound, since deploying LNMs on event-driven hardware would require only active neurons to compute the update. However, in our GPU-based experimental setup, the update for every neuron, whether it receives input or not, must be computed. Approximating the overhead of event-driven hardware remains difficult due to limited availability.

Table 3: Approximate energy consumption for third-degree LNM and LIF models. ResNet-19 was used for CIFAR-10, CIFAR-100, and CIFAR-10 DVS. ResNet-34 was used for ImageNet.

| Dataset | Timesteps | LIF Energy (mJ) | LNM Energy (mJ) | LNM Overhead |
|---|---|---|---|---|
| CIFAR-10 | 2 | 0.547 | 0.570 | 4.20% |
| CIFAR-100 | 2 | 0.649 | 0.672 | 3.54% |
| CIFAR-10 DVS | 10 | 4.758 | 5.015 | 5.40% |
| ImageNet | 4 | 13.725 | 14.020 | 2.15% |

## 6 Limitations

Our choice of low-degree polynomials may limit the expressiveness and variability of learned neuron models. Additionally, clipping the input to polynomials to the range $[-1, 1]$ can cause information loss, as values outside this range become indistinguishable. Therefore, finding alternatives to polynomials that don't significantly increase energy consumption or computational cost, or decrease model stability, is important for future research.

Another limitation of our work is that it only considers single-variable nonlinear neuron models. Multi-variable neuron models can exhibit biologically plausible dynamics, albeit at the cost of increased computational complexity. Exploring how these multivariate neuron models can be incorporated into our LNM methodology to improve performance further remains an interesting future research direction.

Lastly, our formulation of LNMs introduces challenges in identifying optimal surrogate gradient parameters. While Lian et al. (2023) proposes a method to analytically derive optimal surrogate gradient windows for LIF neurons based on the statistical properties of the neuron, extending this approach to LNMs is non-trivial.

## 7 CONCLUSION

In this work, we introduced Learnable Neuron Models (LNMs) as a scalable and efficient solution to the limitations of traditional neuron models in deep SNNs. By learning expressive, non-linear dynamics directly from data, LNMs enable more adaptable and high-performing spiking architectures without requiring manual tuning or sacrificing training stability. LNMs demonstrate state-of-the-art performance for convolutional SNNs on several benchmark datasets, including CIFAR-10, CIFAR-100, ImageNet, and CIFAR-10 DVS, showcasing their effectiveness as an alternative to traditionally used neuron models.

## 8 REPRODUCIBILITY STATEMENT

To reproduce our results, one can refer to the following information. Appendix A details the dataset augmentations used, training setup, and hyperparameters for each experiment. Additionally, we have included our code repository with this submission. Each experiment has a dedicated Slurm script, which details the exact system setup used. We also include a requirements file within this repository to facilitate the easy recreation of our experimental environment.

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

## A   HYPERPARAMETERS AND EXPERIMENTAL SETUP

Table 4 details the hyperparameters used for each dataset and experiment. Within our included code artifact, all scripts used to run each experiment are provided, enabling future researchers to easily reproduce our experimental setup. Additionally, these scripts detail the hardware setup and the time allotted for each experiment. Lastly, the same random number generation seeds are used across all configurations and are detailed in the code artifact.

| Dataset | CIFAR-10 | CIFAR-100 | CIFAR-10 DVS | ImageNet |
|---|---|---|---|---|
| Architecture | ResNet-19 | ResNet-19 | VGGSNN/ResNet-19 | ResNet-34 |
| Timesteps | 2/4 | 2/4 | 10 | 4 |
| Surrogate Gradient | Rectangle | Rectangle | Triangle | Rectangle |
| Surrogate Gradient Window | [-0.5, 0.5] | [-0.5, 0.5] | [-0.5, 0.5] | [-0.5, 0.5] |
| Epochs | 400 | 400 | 200 | 350 |
| Optimizer | SGD | SGD | SGD | SGD |
| Scheduler | Cosine | Cosine | Cosine | Cosine |
| Learning Rate | 1e-1 | 1e-1 | 1e-1 | 1e-0 |
| LNM Learning Rate | 1e-2 | 1e-2 | 1e-3 | 1e-2 |
| Momentum | 9e-1 | 9e-1 | 9e-1 | 9e-1 |
| Weight Decay | 5e-4 | 5e-4 | 5e-4 | 5e-5 |
| Label Smoothing | 1e-1 | 1e-1 | 1e-2 | 0e-0 |
| Warm Up Epochs | 0 | 0 | 0 | 5 |
| A100 80GB GPUs | 1 | 1 | 1 | 4 |
| Initial Decay | 0.25 | 0.25 | 0.25 | 0.25 |
| Threshold | 0.5 | 0.5 | 0.5 | 0.5 |

Table 4: Hyperparameters for each dataset and model architecture.

### A.1   DATASET AUGMENTATIONS

We detail the data augmentations applied to each dataset used within our work. Note that for all datasets besides CIFAR-10 DVS, we normalize by the mean and standard deviation. For CIFAR-10 (Krizhevsky, 2009), we utilize cropping, flipping, and Cutout (DeVries & Taylor, 2017). For CIFAR-100 (Krizhevsky, 2009), we similarly utilize cropping, flipping, and Cutout (DeVries & Taylor, 2017) with the addition of AutoAugment (Cubuk et al., 2019). For CIFAR-10 DVS Li et al. (2017), we resize each frame to $48 \times 48$ and apply random cropping and flipping. Finally, for ImageNet Deng et al. (2009), we utilize random cropping and flipping.

## B    PSEUDOCODE

Here we present pseudocode for forward and backwards propagation using the LNM and LIF neuron models.

---

**Algorithm 1** LNM: Learnable Neuron Model (Forward and Backward)

---

1: **Initialize:** parameterized learnable membrane update $f_\theta(v)$
2: thresholds $v_{\text{th}}$, reset value $v_{\text{reset}}$
3: membrane voltage $v \leftarrow v_{\text{reset}}$
4: **function** FORWARD$(I, v)$
5:     $v_{\text{in}} \leftarrow v - f_\theta(v) + I$
6:     $z \leftarrow \Theta(v_{\text{in}} - v_{\text{th}})$
7:     **if** $z = 1$ **then**
8:         $v_{\text{new}} \leftarrow v_{\text{reset}}$
9:     **else**
10:        $v_{\text{new}} \leftarrow v_{\text{in}}$
11:    **end if**
12:    **return** $(z, v_{\text{new}})$
13: **end function**

14: **function** BACKWARD$(\frac{\partial \mathcal{L}}{\partial z}, \frac{\partial \mathcal{L}}{\partial v_{\text{new}}})$
15:    $\frac{\partial \mathcal{L}}{\partial v_{\text{in}}} \leftarrow \frac{\partial \mathcal{L}}{\partial z} \cdot \text{surrogate\_grad}(v_{\text{in}} - v_{\text{th}}) + \frac{\partial \mathcal{L}}{\partial v_{\text{new}}} \cdot \mathbf{1}[z = 0]$
16:    $\frac{\partial \mathcal{L}}{\partial v} \leftarrow \frac{\partial \mathcal{L}}{\partial v_{\text{in}}} \cdot (1 - f'_\theta(v))$
17:    $\frac{\partial \mathcal{L}}{\partial \theta_k} \leftarrow$ Equation 12
18:    **return** gradients w.r.t. $v$ and $f$
19: **end function**

---

---

**Algorithm 2** LIF: Leaky Integrate-and-Fire Neuron (Forward and Backward)

---

1: **Initialize:** leak constant $\tau_{\text{inv}}$
2: thresholds $v_{\text{th}}$, reset value $v_{\text{reset}}$
3: membrane voltage $v \leftarrow v_{\text{reset}}$
4: **function** FORWARD$(I, v)$
5:     $v_{\text{in}} \leftarrow v - \tau_{\text{inv}} v + I$
6:     $z \leftarrow \Theta(v_{\text{in}} - v_{\text{th}})$
7:     **if** $z = 1$ **then**
8:         $v_{\text{new}} \leftarrow v_{\text{reset}}$
9:     **else**
10:        $v_{\text{new}} \leftarrow v_{\text{in}}$
11:    **end if**
12:    **return** $(z, v_{\text{new}})$
13: **end function**

14: **function** BACKWARD$(\frac{\partial \mathcal{L}}{\partial z}, \frac{\partial \mathcal{L}}{\partial v_{\text{new}}})$
15:    $\frac{\partial \mathcal{L}}{\partial v_{\text{in}}} \leftarrow \frac{\partial \mathcal{L}}{\partial z} \cdot \text{surrogate\_grad}(v_{\text{in}} - v_{\text{th}}) + \frac{\partial \mathcal{L}}{\partial v_{\text{new}}} \cdot \mathbf{1}[z = 0]$
16:    $\frac{\partial \mathcal{L}}{\partial v} \leftarrow \frac{\partial \mathcal{L}}{\partial v_{\text{in}}} \cdot (1 - \tau_{\text{inv}})$
17:    **return** gradient w.r.t. $v$
18: **end function**

---

# C    ABLATION STUDY ON BOUNDING METHODS

In this study, we investigate the effect of different bounding methods on the performance of our low-degree polynomial neuron models. Table 5 summarizes the results on CIFAR-100 using a ResNet-19 architecture with 4 timesteps.

| Dataset | Bounding Method | Architecture | Timesteps | Accuracy |
|---------|-----------------|--------------|-----------|----------|
| CIFAR-100 | TanH | ResNet-19 | 4 | $80.52 \pm 0.08\%$ |
| CIFAR-100 | Clip [-5, 5] | ResNet-19 | 4 | $80.61 \pm 0.09\%$ |
| CIFAR-100 | Clip [-1, 1] | ResNet-19 | 4 | $\mathbf{80.70 \pm 0.14\%}$ |

Table 5: Effect of different bounding methods on model accuracy. The clip range [-1,1] yields the best performance for low-degree polynomial neurons.

We observe that the choice of bounding method has a notable impact on model performance. In particular, expanding the clip range tends to slightly reduce accuracy. This behavior can be attributed to the limited expressivity of low-degree polynomial neuron models. That is, constraining the input domain keeps the polynomial within a region where it can more effectively approximate the target nonlinear dynamics. When the input range is widened, the polynomial must represent a larger portion of the voltage space, which increases approximation error and degrades performance.

In the TanH bounding method, the function's inherent smoothness suppresses abrupt voltage transitions. While this stabilizes the membrane dynamics, it may also reduce the neuron's sensitivity to salient input features. In contrast, clipping the membrane potential to the range $[-1, 1]$ allows sharper transitions within the neuron's critical voltage region, which appears to enhance the neuron's ability to discriminate informative inputs, thereby improving overall accuracy.

