# OpenReview forum: "Learning Neuron Dynamics within Deep Spiking Neural Networks"
_ICLR.cc/2026/Conference — Submitted to ICLR 2026_

### Official Review · Reviewer_ZGV9 · 2025-10-27

**Soundness:** 2
**Presentation:** 3
**Contribution:** 2
**Rating:** 4
**Confidence:** 4

**Summary:**

This paper proposes a new neuron formulation for spiking neural networks (SNNs), called the Learnable Neuron Model (LNM). Unlike conventional Leaky Integrate-and-Fire (LIF) or other fixed neuron models, LNM allows the dynamics of each neuron layer to be learned directly from data during training. The authors instantiate this idea using low-degree polynomial parameterizations, trained via backpropagation with surrogate gradients. The approach demonstrates superior accuracy across multiple datasets, including CIFAR-10, CIFAR-100, CIFAR10-DVS, and ImageNet, with only a minor energy overhead.

**Strengths:**

1. The paper introduces a parametric family of nonlinear integrate-and-fire models that unifies existing neuron formulations under a learnable framework.

2. Experiments on standard datasets demonstrate the superior performance of LNM.

**Weaknesses:**

1. While the authors visualize diverse learned neuron behaviors, there is no deeper analysis of how these dynamics contribute to improved performance and why certain layers benefit from non-LIF behavior.

2. The model restricts f_θ(u) to low-degree polynomials and clips inputs to ensure stability, yet this also limits expressiveness. The impact of this design choice on the ability to model long-term temporal dependencies remains unclear.

3. The paper focuses exclusively on convolutional SNNs. It does not investigate whether LNM dynamics generalize to spiking transformers or temporal sequence models.

4. The paper does not compare existing spiking neuron models.

**Questions:**

1. How does the choice of polynomial degree interact with the number of time steps?

2. Could the authors conduct ablation studies to compare LNM against those fixed neuron models?

3. How would LNM behave on temporal datasets (e.g., PSMNIST, Sequential CIFAR10), where temporal credit assignment is crucial?

4. Could the authors clarify whether the learned neuron parameters generalize across architectures or must be retrained from scratch?

---

> ### Author Response · Authors · 2025-11-18
> **Response 1**
>
> Dear Reviewer ZGV9,
>
> Thank you for your in-depth summary, questions, and concerns on our work. We hope the following responses help to address them.
>
> # Could the authors conduct ablation studies to compare LNM against those fixed neuron models?
>
> Thank you for pointing out the lack of comparison with fixed-neuron models. The LIF and Integrate-and-Fire (IF) are the only two fixed neuron models we are aware of that have been successfully applied to high-performance deep learning settings. As the LIF neuron nearly always outperforms the IF neuron, we have performed several new experiments using the LIF neuron under the exact same experimental setup as the LNM neuron model. That is, the LIF neuron has the same decay, threshold, and surrogate gradient window as the LNM neuron model. The following table compares the results of both these neuron models, CIFAR-10, CIFAR-100, and ImageNet. Similarly, we used ResNet-19 for CIFAR-10/CIFAR-100 and ResNet-34 for ImageNet.
>
> # The paper focuses exclusively on convolutional SNNs. It does not investigate whether LNM dynamics generalize to spiking transformers or temporal sequence models.
>
> Full-scale transformer experiments are not included in this rebuttal because we are not yet confident in our preliminary results and prefer not to present analyses that were rushed or insufficiently validated. Importantly, the ideology behind LNMs is architecture-agnostic, and our initial checks suggest that integrating it into transformer-based models requires minimal modification to their core architecture. We view a thorough investigation of applying LNM to transformer settings as a natural and valuable direction for future work. At this stage, we believe the presented results already demonstrate a meaningful and incremental contribution over prior research without relying on incomplete transformer experiments.
>
>
> # Could the authors clarify whether the learned neuron parameters generalize across architectures or must be retrained from scratch?
>
> Thank you for raising this question. Our current results suggest that learned neuron parameters can transfer across datasets when the underlying model architecture is held fixed. That is, there is no constraint that prevents the reuse of learned neuron models in this setting. We suspect the model's weights will adapt to the neuron models, though there is no guarantee of superior performance. However, fine-tuning the neuron models may alleviate this issue.
>
> Transferring neuron parameters across architectures, however, is less straightforward. As shown in Figure 2, different layers within the same architecture already learn distinct neuron models. This layer-specific specialization makes it unclear how one should select or aggregate neuron models when moving to a structurally different architecture. We have not yet identified a principled method for doing so.
>
> Overall, while cross-architecture reuse remains an open challenge, we agree it presents an interesting direction for future work. Specifically, the possibility of learning a more universal neuron model that generalizes across diverse SNN architectures.
>
> Therefore, at the moment, we feel it is best to retrain neuron models from scratch if performance is the overarching goal. However, fine-tuning neuron models across different datasets with the same architecture may be an exception to this.

---

> ### Author Response · Authors · 2025-11-18
> **Response 2**
>
> # Could the authors conduct ablation studies to compare LNM against those fixed neuron models?
>
> Thank you for pointing out the lack of comparison with fixed-neuron models. The LIF and Integrate-and-Fire (IF) are the only two fixed neuron models we are aware of that have been successfully applied to high-performance deep learning settings. As the LIF neuron nearly always outperforms the IF neuron, we have performed several new experiments using the LIF neuron under the exact same experimental setup as the LNM neuron model. That is, the LIF neuron has the same decay, threshold, and surrogate gradient window as the LNM neuron model. The following table compares the results of both these neuron models, CIFAR-10, CIFAR-100, and ImageNet. Similarly, we used ResNet-19 for CIFAR-10/CIFAR-100 and ResNet-34 for ImageNet.
>
> | **Neuron Model**                  | **Timesteps**   | **CIFAR-10**              | **CIFAR-100**             | **ImageNet**              |
> |---------------------------|------------------|---------------------------|---------------------------|---------------------------|
> | LIF (Our Implementation)  | 4/4/4            | 96.20 ± 0.20%             | 78.95 ± 0.02%             | 69.04%                    |
> | LIF (Our Implementation)  | 2/2/×           | 96.16 ± 0.01%             | 79.92 ± 0.10%             | ×                         |
> | **LNM**            | **4/4/4**        | **97.01 ± 0.04%**         | **80.70 ± 0.14%**         | **70.91% ± 0.03%**        |
> | **LNM**            | **2/2/×**       | **96.96 ± 0.10%**         | **80.07 ± 0.13%**         | ×                         |
>
> On CIFAR-10, LNM outperforms LIF by 0.8% and 0.81% with two and four timesteps, respectively. On CIFAR-100, LNM achieves 0.15% higher accuracy with two timesteps and 1.75% higher accuracy with four timesteps, compared to LIF. Finally, on ImageNet, LNM outperforms LIF by a substantial 1.87%.
>
> Next, let's compare the two neuron models on CIFAR-10 DVS. Looking at the following table, we see that with VGGSNN, LNM outperforms LIF by 7.85%. LNM also outperforms LIF on ResNet-19 by 8.23%. While these differences initially seemed too large, they are in line with the reported values shown in [1], [2], and [3].
>
> | **Dataset**| **Neuron Model**                 | **Architecture** | **Timesteps** | **Accuracy**           |
> |-------------|---------------------------|-------------------|---------------|-------------------------|
> | CIFAR-10 DVS    | LIF (Our Implementation)  | VGGSNN            | 10            | 75.10 ± 1.31%          |
> | CIFAR-10 DVS   | LIF (Our Implementation)  | ResNet-19         | 10            | 73.16 ± 1.37%          |
> | CIFAR-10 DVS   | **LNM**                   | **VGGSNN**        | **10**        | **82.95 ± 0.69%**      |
> | CIFAR-10 DVS   | **LNM**            | **ResNet-19**     | **10**        | **81.39 ± 0.25%**      |
>
> We have added these new results to Tables 1 and 2 of our work, which we feel have strengthened the performance claims of LNMs over LIF.
>
> [1] Bhaskar Mukhoty, Velibor Bojkovic, William de Vazelhes, Xiaohan Zhao, Giulia De Masi, Huan Xiong, and Bin Gu. Direct training of SNN using local zeroth order method. In Thirty-seventh Conference on Neural Information Processing Systems, 2023.
>
> [2] Yufei Guo, Yuhan Zhang, Yuanpei Chen, Weihang Peng, Xiaode Liu, Liwen Zhang, Xuhui Huang, and Zhe Ma. Membrane potential batch normalization for spiking neural networks. In 2023 IEEE/CVF International Conference on Computer Vision (ICCV), pages 19363–19373, 2023.
>
> [3] Yufei Guo, Yuanpei Chen, Liwen Zhang, Xiaode Liu, Yinglei Wang, Xuhui Huang, and Zhe Ma. Im-loss: Information maximization loss for spiking neural networks. In S. Koyejo, S. Mohamed, A. Agarwal, D. Belgrave, K. Cho, and A. Oh, editors, Advances in Neural Information Processing Systems, volume 35, pages 156–166. Curran Associates, Inc., 2022.

---

> ### Author Response · Authors · 2025-11-19
> **Response 3**
>
> # The model restricts $f_\theta(u)$ to low-degree polynomials and clips inputs to ensure stability, yet this also limits expressiveness. The impact of this design choice on the ability to model long-term temporal dependencies remains unclear.
>
> To assess how different bounding strategies influence the learned neuron dynamics, we conducted a small ablation study on CIFAR-10 DVS. We compare three bounding strategies for LNM: (1) the standard $[-1, 1]$ clipping used throughout this work, (2) a much wider $[-5, 5]$ clipping range, and (3) the tanh function as a smooth, non-linear alternative. The results are shown in the table below.
>
> | Dataset         | Bounding Method | Architecture | Timesteps | Accuracy              |
> |-----------------|-----------------|--------------|-----------|------------------------|
> | CIFAR-10 DVS   | TanH            | ResNet-19    | 10         | 80.90 ± 0.42%          |
> | CIFAR-10 DVS    | Clip [-5, 5]    | ResNet-19    | 10         | 74.79 ± 1.98%          |
> | CIFAR-10 DVS   | Clip [-1, 1]    | ResNet-19    | 10         | **81.39 ± 0.25%**      |
>
> These results show that the choice of bounding method substantially affects model performance. Using a smooth function, such as tanh, results in only moderate degradation, which we suspect arises because it suppresses sharp voltage transitions and dampens the neuron's responsiveness to informative temporal features. In contrast, widening the clipping range to $[-5, 5]$ is detrimental. That is, the bounds are too large, leading to numerical instability that significantly degrades accuracy. Overall, while clipping may restrict expressiveness, these results indicate that it is necessary and that tightly clipping outliers is preferable to smoothing the overall shape of the update function.

---

> > ### Author Response · Authors · 2025-11-19
> > **Response 4**
> >
> > # While the authors visualize diverse learned neuron behaviors, there is no deeper analysis of how these dynamics contribute to improved performance and why certain layers benefit from non-LIF behavior.
> >
> > Our study contributes to the broader line of work exploring non-linear spiking neuron models that balance computational complexity with performance benefits. Prior research has proposed alternatives or modifications to LIF, including [1], [2], [3], [4], and others. Motivated by similar questions about whether richer neuron dynamics can improve learning. Our work follows this same direction by systematically examining how much non-linearity can be introduced into a neuron model while remaining both computationally tractable and energy-efficient.
> >
> > A key objective of the paper is to investigate whether controlled non-linearity in the membrane update can help SNNs learn more expressive temporal features. This is reflected in two experiments: (1) evaluating performance as we vary the polynomial degree vs LIF (Figure 1). (2) Comparing the different learned update rules and their performance relative to recent works across a variety of models and datasets. Across all of these studies, we observe that the learned non-linear models consistently outperform the LIF-based results. These findings collectively suggest that non-linear dynamics meaningfully expand the representational capacity of the neuron model.
> >
> > At the same time, our work does not attempt a layer-by-layer causal analysis of why specific learned dynamics lead to improvements or why certain layers deviate more strongly from LIF behavior. Performing such a study would require careful attribution methods tailored to spiking systems, for which we currently lack a suitable methodology. Instead, our results provide empirical support for the broader claim: allowing the neuron to learn non-linear update rules leads to consistent performance gains across architectures and datasets.
> >
> > We agree that a deeper analysis of where and why non-LIF behavior is most beneficial represents an interesting direction for future work. In particular, exploring hybrid architectures in which certain layers adopt learned non-linear neuron models while others remain LIF may reveal whether different network stages benefit from distinct dynamical regimes. While such an investigation lies beyond the scope of our current study, our positive empirical results indicate that non-linearity is a promising avenue for improving SNN performance.
> >
> > [1] Shibo Feng, Wanjin Feng, Xingyu Gao, Peilin Zhao, and Zhiqi Shen. Ts-lif: A temporal segment spiking neuron
> > network for time series forecasting, 2025.
> >
> > [2] Yufei Guo, Yuanpei Chen, Xiaode Liu, Weihang Peng, Yuhan Zhang, Xuhui Huang, and Zhe Ma. Ternary
> > spike: learning ternary spikes for spiking neural networks. In Proceedings of the Thirty-Eighth AAAI Conference
> > on Artificial Intelligence and Thirty-Sixth Conference on Innovative Applications of Artificial Intelligence and
> > Fourteenth Symposium on Educational Advances in Artificial Intelligence, AAAI’24/IAAI’24/EAAI’24. AAAI
> > Press, 2024.
> >
> > [3] Shuang Lian, Jiangrong Shen, Ziming Wang, and Huajin Tang. Im-lif: Improved neuronal dynamics with
> > attention mechanism for direct training deep spiking neural network. IEEE Transactions on Emerging Topics
> > in Computational Intelligence, 8(2):2075–2085, 2024.
> >
> > [4] Shimin Zhang, Qu Yang, Chenxiang Ma, Jibin Wu, Haizhou Li, and Kay Chen Tan. Tc-lif: A two-compartment
> > spiking neuron model for long-term sequential modelling, 2024.

---

> > > ### Author Response · Authors · 2025-11-23
> > > **Response 5**
> > >
> > > # How does the choice of polynomial degree interact with the number of time steps?
> > >
> > > In this ablation study, we examine how the polynomial degree and number of timesteps jointly affect model performance. Using CIFAR-100, we sweep polynomial degrees in ${1, 3, 5, 7}$ and timesteps in ${2, 4, 6}$, and report the mean and standard deviation of validation accuracy over three runs. The full results are shown in the following table.
> > >
> > > | **Polynomial Degree** | **2 Timesteps**        | **4 Timesteps**        | **6 Timesteps**        |
> > > |-----------------------|------------------------|-------------------------|-------------------------|
> > > | 1                     | 80.43 ± 0.21%          | 80.52 ± 0.06%           | 77.78 ± 0.39%           |
> > > | 3                     | 80.28 ± 0.27%          | 80.70 ± 0.17%           | 80.34 ± 0.17%           |
> > > | 5                     | 80.40 ± 0.15%          | 80.70 ± 0.22%           | 79.96 ± 1.19%           |
> > > | 7                     | 80.23 ± 0.13%          | 80.71 ± 0.13%           | 80.44 ± 0.15%           |
> > >
> > >
> > > At two timesteps, the results indicate that increasing the polynomial degree does not yield consistent improvements; instead, performance remains similar across all polynomial degrees. At four and six timesteps, there is a mild trend toward better accuracy with higher degrees, though the differences remain small and largely within overlapping standard deviations. This suggests that higher-degree polynomials may provide limited benefits at longer temporal horizons, but these gains come with increased computational cost and a higher risk of training instability. Consequently, while higher-degree polynomials can help in some settings, their advantages should be weighed against these practical drawbacks.
> > >
> > > We have added this discussion to Appendix D.

---

> > > > ### Author Response · Authors · 2025-11-23
> > > > **Response 5**
> > > >
> > > > # How would LNM behave on temporal datasets (e.g., PSMNIST, Sequential CIFAR10), where temporal credit assignment is crucial?
> > > >
> > > > To evaluate LNM on a temporal task, we experiment with psMNIST [1], using the same model setup as in [2] to capture long-term dependencies. Results are reported in the following table.
> > > >
> > > > | **Dataset** | **Neuron Model** | **Accuracy**           |
> > > > |-------------|-----------------|-----------------------|
> > > > | psMNIST     | LIF             | 96.66 ± 0.085%        |
> > > > | psMNIST     | LNM             | 97.21 ± 0.143%        |
> > > >
> > > > On this task, LNM achieves a 0.55\% improvement over LIF. Given the importance of temporal credit assignment in psMNIST, this result indicates that LNMs can effectively learn neuron dynamics that better capture temporal dependencies compared to LIF.
> > > >
> > > > [1] Geoffrey Hinton, Oriol Vinyals, and Jeff Dean. Distilling the knowledge in a neural network, 2015.
> > > >
> > > > [2] Wei Fang, Zhaofei Yu, Zhaokun Zhou, Ding Chen, Yanqi Chen, Zhengyu Ma, Timoth´ee Masquelier, and
> > > > Yonghong Tian. Parallel spiking neurons with high efficiency and ability to learn long-term dependencies. In
> > > > Thirty-seventh Conference on Neural Information Processing Systems, 2023.

---

> > > > > ### Comment · Reviewer_ZGV9 · 2025-11-25
> > > > >
> > > > > Thank you for providing additional experiments, and I am convinced that LNM outperforms the LIF model in certain cases. However, the previously noted weaknesses, including the lack of in-depth analysis (W1), the limited evaluation on temporal datasets (Q3), and the lack of comparisons with more recently proposed spiking neuron models (W4), still remain insufficiently addressed. The very small accuracy changes for different polynomial degrees raise further concerns about whether this component is truly necessary or provides meaningful benefit. I am also not convinced by the results on psMNIST. In particular, the reported accuracy of the LIF model appears unexpectedly high compared to typical results in the literature. Given these remaining concerns, I am keeping my original score.

---

> > > > > > ### Author Response · Authors · 2025-11-25
> > > > > >
> > > > > > We sincerely thank you for the time you devoted to reviewing our work, engaging with our rebuttal, and providing additional feedback. We appreciate your acknowledgement of the strengths of our approach and respect your remaining concerns and score decision.

---

### Official Review · Reviewer_TxX4 · 2025-10-29

**Soundness:** 3
**Presentation:** 3
**Contribution:** 3
**Rating:** 6
**Confidence:** 5

**Summary:**

The authors use spiking neural networks (SNN) on several vision datasets (CIFAR10, CIFAR100, ImageNet, and CIFAR10-DVS).

While most of the SNN community uses leaky integrate-and-fire (LIF) neurons, here the authors propose to learn more complex spiking neuron models using backprop. Indeed, the function f(u), which defines the neuronal dynamics, and which is equal to -u for a LIF neuron, is defined as a 3rd order polynomial whose coefficients are learned.

**Strengths:**

* To the best of my knowledge, this idea is new.
* Using it, they improve the SOTA on CIFAR10, CIFAR100, and ImageNet.

**Weaknesses:**

* The authors need to show that the learnable dynamics outperforms the LIF neuron, using exactly the same architecture, learning procedure, etc.
For example, the LIF accuracy + std could be added to Fig. 1.

* On CIFAR10-DVS this paper reached a better accuracy (85.90% with T=10):
Fang W et al. (2023) Parallel Spiking Neurons with High Efficiency and Ability to Learn Long-term Dependencies. NeurIPS 1:1–13 Available at: http://arxiv.org/abs/2304.12760.
It should be added to Table 2.

* The energy estimation ignores memory accesses, whereas they dominate the budget!
See:

Lemaire, E., Cordone, L., Castagnetti, A., Novac, P. E., Courtois, J., & Miramond, B. (2022, November). An analytical estimation of spiking neural networks energy efficiency. In International Conference on Neural Information Processing (pp. 574-587). Cham: Springer International Publishing.

Dampfhoffer, M., Mesquida, T., Valentian, A., & Anghel, L. (2022). Are SNNs really more energy-efficient than ANNs? an in-depth hardware-aware study. IEEE Transactions on Emerging Topics in Computational Intelligence.

MINOR POINTS:

* Eq 4. The authors could precise that this corresponds to a "hard reset".

* p6: "However, we also observe increased variability as the degree increases."
This is not clear. The highest variability is for 2nd degree.

**Questions:**

Would it make sense to include the input current I as an argument of the learnable f function in Eq. 2?

---

> ### Author Response · Authors · 2025-11-18
> **Response 1**
>
> Dear Reviewer TxX4,
>
> We thank you for your comments and questions on our work as well as acknowledging its novelty. We hope the following responses help to clarify your concerns.
>
> # The authors need to show that the learnable dynamics outperforms the LIF neuron, using exactly the same architecture, learning procedure, etc. For example, the LIF accuracy + std could be added to Fig. 1.
>
> Thank you for pointing out this lack of comparison with the LIF neuron. To remedy this, we have performed several new experiments using the LIF neuron under the exact same experimental setup as the LNM neuron model. That is, the LIF neuron has the same decay, threshold, and surrogate gradient window as the LNM neuron model. The following table compares the results of both these neuron models, CIFAR-10, CIFAR-100, and ImageNet. Similarly, we used ResNet-19 for CIFAR-10/CIFAR-100 and ResNet-34 for ImageNet.
>
> | **Neuron Model**                  | **Timesteps**   | **CIFAR-10**              | **CIFAR-100**             | **ImageNet**              |
> |---------------------------|------------------|---------------------------|---------------------------|---------------------------|
> | LIF (Our Implementation)  | 4/4/4            | 96.20 ± 0.20%             | 78.95 ± 0.02%             | 69.04%                    |
> | LIF (Our Implementation)  | 2/2/×           | 96.16 ± 0.01%             | 79.92 ± 0.10%             | ×                         |
> | **LNM**            | **4/4/4**        | **97.01 ± 0.04%**         | **80.70 ± 0.14%**         | **70.91% ± 0.03%**        |
> | **LNM**            | **2/2/×**       | **96.96 ± 0.10%**         | **80.07 ± 0.13%**         | ×                         |
>
> On CIFAR-10, LNM outperforms LIF by 0.8% and 0.81% with two and four timesteps, respectively. On CIFAR-100, LNM achieves 0.15% higher accuracy with two timesteps and 1.75% higher accuracy with four timesteps, compared to LIF. Finally, on ImageNet, LNM outperforms LIF by a substantial 1.87%.
>
> Next, let's compare the two neuron models on CIFAR-10 DVS. Looking at the following table, we see that with VGGSNN, LNM outperforms LIF by 7.85%. LNM also outperforms LIF on ResNet-19 by 8.23%. While these differences initially seemed too large, they are in line with the reported values shown in [1], [2], and [3].
>
> | **Dataset**| **Neuron Model**                 | **Architecture** | **Timesteps** | **Accuracy**           |
> |-------------|---------------------------|-------------------|---------------|-------------------------|
> | CIFAR-10 DVS    | LIF (Our Implementation)  | VGGSNN            | 10            | 75.10 ± 1.31%          |
> | CIFAR-10 DVS   | LIF (Our Implementation)  | ResNet-19         | 10            | 73.16 ± 1.37%          |
> | CIFAR-10 DVS   | **LNM**                   | **VGGSNN**        | **10**        | **82.95 ± 0.69%**      |
> | CIFAR-10 DVS   | **LNM**            | **ResNet-19**     | **10**        | **81.39 ± 0.25%**      |
>
> We have added these new results to Tables 1 and 2 of our work, which we feel have strengthened the performance claims of LNMs over LIF.
>
> [1] Bhaskar Mukhoty, Velibor Bojkovic, William de Vazelhes, Xiaohan Zhao, Giulia De Masi, Huan Xiong, and Bin Gu. Direct training of SNN using local zeroth order method. In Thirty-seventh Conference on Neural Information Processing Systems, 2023.
>
> [2] Yufei Guo, Yuhan Zhang, Yuanpei Chen, Weihang Peng, Xiaode Liu, Liwen Zhang, Xuhui Huang, and Zhe Ma. Membrane potential batch normalization for spiking neural networks. In 2023 IEEE/CVF International Conference on Computer Vision (ICCV), pages 19363–19373, 2023.
>
> [3] Yufei Guo, Yuanpei Chen, Liwen Zhang, Xiaode Liu, Yinglei Wang, Xuhui Huang, and Zhe Ma. Im-loss: Information maximization loss for spiking neural networks. In S. Koyejo, S. Mohamed, A. Agarwal, D. Belgrave, K. Cho, and A. Oh, editors, Advances in Neural Information Processing Systems, volume 35, pages 156–166. Curran Associates, Inc., 2022.

---

> > ### Author Response · Authors · 2025-11-18
> > **Response 2**
> >
> > # On CIFAR10-DVS, this paper reached a better accuracy (85.90\% with T=10): Fang W et al. (2023) Parallel Spiking Neurons with High Efficiency and Ability to Learn Long-term Dependencies. NeurIPS 1:1–13 Available at: http://arxiv.org/abs/2304.12760. It should be added to Table 2.
> >
> > Thank you for pointing out this missing comparison in our work. We have updated Table 2 and the corresponding discussion accordingly.
> >
> > # The energy estimation ignores memory accesses, whereas they dominate the budget!
> >
> > Thank you for these resources. We are currently using them to update our energy estimation to include memory accesses. We will respond once more once we have our new results.
> >
> > # Would it make sense to include the input current I as an argument of the learnable f function in Eq. 2?
> >
> > This is an excellent question, and we spent time investigating it. While the input current $I$ is not typically included inside the function
> > $f$ for general non-linear integrate-and-fire neuron models, it is indeed interesting to consider what would happen if it were. Intuitively, incorporating $I$ directly into $f$ could help the neuron learn correlations between the input current and the membrane voltage, potentially enabling richer dynamics and improved performance. However, when considering how to include $I$ in $f$, we encounter two major issues.
> >
> > First, in our current setup, we learn a single neuron model per layer; this is analogous to how LIF layers typically share decay and threshold parameters. This design keeps both memory and training costs manageable. If $f$ were to depend on $I$, it would be difficult to justify sharing the same $f$ across all neurons in the layer, since each neuron receives a different input current, which intuitively relates to different learned features. Making $f$ unique for every neuron would dramatically increase the parameter count and training complexity. In future work, we would still like to attempt to train a model with a unique $f$ per neuron while also incorporating $I$ to see if there is a significant performance improvement that outweighs the memory and computational overhead.
> >
> > Second, if we were to keep a shared $f$ per layer while somehow incorporating $I$, we would run into the problem that each neuron's input current would implicitly influence the dynamics of every other neuron in the layer. This cross-neuron interference introduces noise that we expect would hinder, rather than help, learning.
> >
> > For these reasons, although incorporating $I$ into $f$ is a conceptually interesting direction that may enable more expressive neuronal dynamics, it presents practical challenges that require additional research to address.
> >
> > # Minor Points
> >
> > ## Eq 4. The authors could be more precise and state that this corresponds to a "hard reset".
> >
> > Thank you, we have clarified this in the draft of our paper.
> >
> > ## p6: "However, we also observe increased variability as the degree increases." This is not clear. The highest variability is for 2nd degree.
> >
> > We apologize for this unclear explanation. In addition to clarifying this discussion, we have included an LIF baseline in our ablation study, as another reviewer suggested. The updated discussion of Figure 1 is now:
> >
> > "To determine the optimal polynomial degree, we conduct an ablation study using a ResNet-19 architecture trained on CIFAR-100 with four timesteps. For each polynomial degree, we perform three independent training runs and report the mean and standard deviation to assess stability and performance. Additionally, we include a baseline of the LIF neuron's performance under the same training conditions. As shown in Figure 1, our learnable neuron model does benefit from higher-degree polynomials. However, beyond third-degree polynomials, the mean validation accuracy starts to plateau, and the standard deviation overlaps with higher-degree polynomials. Notably, while second-degree polynomials achieve slightly higher mean accuracy, they exhibit the largest variability, indicating the least stable performance. Consequently, we select third-degree polynomials as the optimal choice, offering a favorable trade-off between accuracy, stability, and computational efficiency."

---

> > > ### Comment · Reviewer_TxX4 · 2025-11-19
> > > **about my question (including I as an argument for f)**
> > >
> > > My idea was to learn a single f(u,I) function per layer.
> > >
> > > >  If $f$ were to depend on $I$, it would be difficult to justify sharing the same $f$ across all neurons in the layer, since each neuron receives a different input current
> > >
> > > I don't see the pb.
> > > $u$ is also neuron dependent, and still you showed that sharing the same $f(u)$ across all neurons in the layer is beneficial.
> > >
> > > > Second, if we were to keep a shared $f$ per layer while somehow incorporating $I$, we would run into the problem that each neuron's input current would implicitly influence the dynamics of every other neuron in the layer.
> > >
> > > I disagree.
> > > The local $I$ would only influence the local $u$.
> > > Exactly like with your current $f(u)$ fonction: the local $u$ is only influenced by the local $u$, even though $f$ is shared.

---

> > > > ### Author Response · Authors · 2025-11-19
> > > > **Response about including I as an argument.**
> > > >
> > > > Thank you for your very prompt response. We see your point much better now.
> > > >
> > > > We had initially interpreted your suggestion as proposing an update rule of the form
> > > >
> > > > $u' = f_\theta(u, I)$,
> > > >
> > > > which would remove $ I$'s independence from the membrane update. However, we now see that you were referring instead to an update rule such as
> > > >
> > > > $u' = f_\theta(u, I) + I$.
> > > >
> > > >
> > > > ## Further Analysis
> > > >
> > > > To better understand what it would mean to learn a function $f(u, I)$, we revisited the structure of our current model, which uses
> > > >
> > > > $u' = f_\theta(u) + I$.
> > > >
> > > > Here, the input current $I$ is defined as:
> > > >
> > > > $I = \sum_i^k w_i \times o_{i}$,
> > > >
> > > > where $w_i$ is the synaptic weight and $o_{i}$ is an output spike from neuron $i$ in the previous layer of the network. (We are simply expanding standard operations such as convolutions or fully connected layers.)
> > > >
> > > > Now let's assume we are using a third-degree LNM and that three neurons in the previous layer are connected to the neuron in question. Then, expanding both $f_\theta(u)$ and $I$ gives us:
> > > >
> > > > $u' = \theta_1 u^1 + \theta_2 u^2 + \theta_3 u^3 + w_1 o_{1} + w_2 o_{2} + w_3 o_{3}$.
> > > >
> > > > Since each term already has its own independent learnable parameter, adding additional learnable weights directly to $I$ or to individual $o_{i}$ terms appears counterintuitive. A more natural way to incorporate $I$ into the learned dynamics may be through interaction terms between $u$ and $I$.
> > > >
> > > > Conceptually, we feel this makes sense. For example, adding the interaction term, $\theta_4 \times u \times I$, could indicate to the neuron when there are major differences between the input current and the voltage, and allow the neuron to learn unique behavior for this situation
> > > >
> > > > This opens several interesting research questions: Which interaction terms should be included? How many? Are interaction terms the best mechanism? Should they operate on the aggregate current $I$ or on individual presynaptic contributions $o_{i}$?
> > > >
> > > > We note that incorporating such terms means the resulting neuron model no longer fits neatly within the standard non-linear integrate-and-fire family. Rather than being a limitation, this simply reflects that the formulation defines a distinct class of neuron models.
> > > >
> > > > Overall, we agree that this is an intriguing direction with many open questions. We appreciate the opportunity to discuss it with you and plan to explore it further.

---

> > > > > ### Comment · Reviewer_TxX4 · 2025-11-20
> > > > >
> > > > > > We note that incorporating such terms means the resulting neuron model no longer fits neatly within the standard non-linear integrate-and-fire family.
> > > > >
> > > > > I agree.
> > > > >
> > > > > > Rather than being a limitation, this simply reflects that the formulation defines a distinct class of neuron models.
> > > > >
> > > > > Yes, it's a limitation because the set of f(u) functions is a subset of the set of f(u,I) functions.
> > > > > However, it's entirely acceptable to restrict the scope of this paper to the standard non-linear integrate-and-fire family and leave the more general formulation for future work.
> > > > >
> > > > > All my other concerns have been addressed.
> > > > >
> > > > > So I have raised my rating to 8.

---

> > > > > > ### Author Response · Authors · 2025-11-20
> > > > > >
> > > > > > We would like to express our sincere gratitude for your decision to raise your score. We enjoyed the discussion and appreciate your quick responses.

---

### Official Review · Reviewer_WbvZ · 2025-11-01

**Soundness:** 3
**Presentation:** 2
**Contribution:** 3
**Rating:** 4
**Confidence:** 4

**Summary:**

This paper introduces Learnable Neuron Models (LNMs) for deep Spiking Neural Networks (SNNs), which parameterize neuron dynamics via low-degree polynomials to learn adaptive non-linear behaviors from data, achieving state-of-the-art performance on multiple datasets with minimal energy overhead compared to traditional LIF models.

**Strengths:**

1. The paper’s proposed Learnable Neuron Models (LNMs) effectively address the limitations of fixed neuron models (e.g., LIF) in deep SNNs by parameterizing neuron dynamics with low-degree polynomials.

2. LNMs exhibit strong flexibility in learning diverse neuron dynamics and align with biological plausibility.

3. The proposed LNMs consistently outperform existing state-of-the-art methods across both static and neuromorphic datasets.

4. This paper is simple and effective, allowing readers to easily identify its innovative points.

**Weaknesses:**

1.  There are multiple writing irregularities violating academic norms: the Abstract has improper line breaks, transitions are abrupt without smooth bridging, and repeated minor errors like "Equation equation" exist in formula references.

2.  The evaluation datasets are insufficient, and a broader range of evaluations should be conducted.

3. The limited innovation in its neuron model design: its core improvement—adopting polynomial functions to parameterize the neuron’s intrinsic dynamic function is essentially a replacement of the learnable parameters in the PLIF model.

**Questions:**

The authors note that complex fixed neuron models introduce extra complexity and hyperparameters, yet their proposed LNMs, which use polynomial parameterization, still introduce "polynomial degree" as a new hyperparameter, failing to fully avoid new hyperparameter introduction.

Please provide the specific calculation process for how this added complexity increases the energy consumption of SNNs trained with LNMs by 2–5.5% compared to the LIF neuron.

---

> ### Author Response · Authors · 2025-11-18
> **Response 1**
>
> Dear Reviewer WbvZ,
>
> We greatly appreciate your comments on our work and thank you for pointing out its clarity in presentation. Below we have detailed responses to your concerns.
>
> # There are multiple writing irregularities violating academic norms: the Abstract has improper line breaks, transitions are abrupt without smooth bridging, and repeated minor errors like "Equation equation" exist in formula references.
>
> We sincerely apologize for these irregularities. We will carefully review our work again to address issues with transitions wherever possible and have removed the line break in the Abstract.
>
> Regarding the “Equation equation” error, in our initial template, which we used to draft the paper, the *eqref* environment did not automatically include the word “Equation,” so we added it manually. When transferring our work to the ICLR template, which automatically includes “Equation,” we did not notice that our manually added "Equation" were now redundant. We apologize for this easily avoidable mistake and appreciate your understanding.
>
> # The limited innovation in its neuron model design: its core improvement—adopting polynomial functions to parameterize the neuron’s intrinsic dynamic function is essentially a replacement of the learnable parameters in the PLIF model.
>
> While the PLIF neuron model was one source of inspiration, the connection is ultimately quite limited. PLIF introduces learnable decay parameters within the LIF formulation, but it remains confined to that specific dynamical form. In contrast, our approach generalizes the idea by parameterizing the entire intrinsic dynamical function of an integrate-and-fire neuron using learnable polynomials. This expands the representable family far beyond LIF. With our formulation, the LNMs can, in principle, learn dynamics corresponding to LIF, QIF, Exponential Integrate-and-Fire, or any other member of the broader non-linear integrate-and-fire family.
>
> When the polynomial reduces to degree 1, our learned neuron model (LNM) closely resembles PLIF. However, even in this restricted case, its behavior is not limited to PLIF’s assumptions. For instance, the LNM can learn negative decay values (even though we do not observe this in practice), while the PLIF can only represent positive decay. As the polynomial degree increases beyond one, the differences become more pronounced. The LNM no longer approximates a singular hand-designed neuron, but instead learns arbitrary nonlinear dynamical responses. Thus, the core innovation is the generalization of learnable dynamics to the entire family of non-linear integrate-and-fire neurons.
>
> # The authors note that complex fixed neuron models introduce extra complexity and hyperparameters, yet their proposed LNMs, which use polynomial parameterization, still introduce "polynomial degree" as a new hyperparameter, failing to fully avoid new hyperparameter introduction.
>
> Thank you for pointing out this unclear section of our paper.
>
> In the following discussion, we will ignore hyperparameters related to the threshold, resting voltage, and surrogate gradient, as these must be considered for all neuron models.
>
> Instead, let's focus on the hyperparameters specific to a particular neuron model. In the case of the LIF neuron, one must only consider the decay value. LNMs, on the other hand, have two hyperparameters, polynomial degree and polynomial initialization. This increases the complexity of the hyperparameter space for LNMs in a non-trivial way. However, as we look at other neuron models in the non-linear integrate-and-fire family, we see that LNMs help to decrease the number of hyperparameters. For instance, the Quadratic Integrate-and-Fire neuron model requires three hyperparameters, the Izhikevich neuron model contains four hyperparameters, and the Adaptive Exponential Integrate-and-Fire neuron model can contain an arbitrary number of hyperparameters (depending on the number of adaptation currents modeled).
>
> Therefore, as we look at the broader family of non-linear integrate-and-fire models, we see that LNMs do indeed help reduce the hyperparameter space. We feel that the increased hyperparameters for these other neuron models are a major reason only Integrate-and-Fire and LIF (along with LIF modifications) have seen success in publications on high-performance spiking neural networks.

---

> > ### Author Response · Authors · 2025-11-18
> > **Response 2**
> >
> > # Please provide the specific calculation process for how this added complexity increases the energy consumption of SNNs trained with LNMs by 2–5.5\% compared to the LIF neuron.
> >
> > We provide detailed calculations of the additional energy consumption introduced by LNMs relative to standard LIF neurons.
> >
> > Membrane update operations.
> > For a single timestep, the LIF membrane update is:
> >
> > $u' = u - (1/\tau) u + I = \beta u + I$
> >
> > where $\tau$ is the membrane time constant and $\beta = 1 - 1/\tau$. This requires 1 multiplication and 1 addition per neuron update.
> >
> > For a degree-$N$ LNM neuron, the membrane update is:
> >
> > $u' = f_\theta(u) + I$
> >
> > where $f_\theta(u)$ is a polynomial of degree $N$ with $\theta_0 = 0$. Computing $f_\theta(u)$ requires $N$ multiplications and $N-1$ additions per neuron update.
> >
> > Adding the input current $I$ introduces $1$ more addition. Therefore, the total per-neuron update cost for a degree-$N$ LNM is $N$ multiplications and $N$ additions
> >
> > Example: Degree-3 LNM.
> > For $N = 3$, this results in 3 multiplications and 3 additions per update, introducing 2 extra multiply-accumulate (MAC) operations compared to the LIF neuron.
> >
> > Using Equation 17 from our work, the energy per layer of the network is
> > $E_i = T * ( fr * E_{AC} * OP_{AC} + E_{MAC} * OP_{MAC} )$
> >
> > where $fr$ is the firing rate, $E_{AC}$ and $E_{MAC}$ are the energy costs of an accumulate and MAC operation, respectively, and $OP_{AC}$ / $OP_{MAC}$ are the corresponding operation counts.
> >
> > Assuming the same spike rate, the energy difference between an LNM and a LIF neuron comes solely from the increased number of MAC operations due to the polynomial update (As AC operations are the synaptic weights). For a degree-3 LNM, this corresponds to a 2–5.5\% increase in energy consumption compared to a LIF neuron in our evaluations.
> >
> > We have updated Section 5.4 (ENERGY EFFICIENCY AND RUNTIME) to help clarify this for future readers.

---

> > > ### Author Response · Authors · 2025-11-25
> > > **Response 3**
> > >
> > > # The evaluation datasets are insufficient, and a broader range of evaluations should be conducted.
> > >
> > > To evaluate LNM on another temporal task, we experiment with psMNIST [1], using the same model setup as in [2] to capture long-term dependencies. Results are reported in the following table.
> > >
> > > | **Dataset** | **Neuron Model** | **Accuracy**           |
> > > |-------------|-----------------|-----------------------|
> > > | psMNIST     | LIF             | 96.66 ± 0.085%        |
> > > | psMNIST     | LNM             | 97.21 ± 0.143%        |
> > >
> > > On this task, LNM achieves a 0.55\% improvement over LIF. Given the importance of temporal credit assignment in psMNIST, this result indicates that LNMs can effectively learn neuron dynamics that better capture temporal dependencies compared to LIF.
> > >
> > > [1] Geoffrey Hinton, Oriol Vinyals, and Jeff Dean. Distilling the knowledge in a neural network, 2015.
> > >
> > > [2] Wei Fang, Zhaofei Yu, Zhaokun Zhou, Ding Chen, Yanqi Chen, Zhengyu Ma, Timoth´ee Masquelier, and
> > > Yonghong Tian. Parallel spiking neurons with high efficiency and ability to learn long-term dependencies. In
> > > Thirty-seventh Conference on Neural Information Processing Systems, 2023.

---

### Official Review · Reviewer_XYhv · 2025-11-01

**Soundness:** 3
**Presentation:** 2
**Contribution:** 2
**Rating:** 2
**Confidence:** 4

**Summary:**

This paper introduces Learnable Neuron Models (LNMs), a parametric formulation for neuron dynamics in Spiking Neural Networks (SNNs) that can be learned directly from data. The authors propose using low-degree polynomials to parameterize the membrane potential dynamics, enabling more expressive temporal behavior than fixed models like Leaky Integrate-and-Fire (LIF). The method is evaluated on multiple static and neuromorphic datasets (CIFAR-10, CIFAR-100, ImageNet, CIFAR-10-DVS) and shows good performance with minimal energy overhead.

**Strengths:**

1. The idea of learning neuron dynamics end-to-end is underexplored in deep SNNs, and this paper presents a clear and practical approach.

2. The paper provides extensive experiments across multiple datasets and architectures, demonstrating consistent improvements over strong baselines.

3. The use of low-degree polynomials and Horner’s method ensures that the approach is computationally efficient and hardware-friendly.

4. The analysis of learned dynamics across layers (e.g., LIF-like, QIF-like, etc.) is insightful and supports the claim that diverse dynamics are beneficial.

**Weaknesses:**

1.  The paper could consider adding pseudocode to more clearly and intuitively demonstrate the differences between the proposed model and LIF.

2. There is a lack of discussion regarding related work on neuron models [1-3].

3. Figure 1 shows that accuracy does not significantly improve as the Polynomial Degree increases.

4. Tables 1 and 2 should include results for the LIF model obtained under the same training conditions and architecture as LNM, to allow for a direct and intuitive comparison of the performance improvement.

5. A main conceptual diagram/figure summarizing the overall method is missing, which would help intuitively illustrate the entire workflow.

[1]  "TS-LIF: A Temporal Segment Spiking Neuron Network for Time Series Forecasting." The Thirteenth International Conference on Learning Representations. 2025.

[2] "CLIF: Complementary Leaky Integrate-and-Fire Neuron for Spiking Neural Networks." International Conference on Machine Learning. PMLR, 2024.

[3]  "Tc-lif: A two-compartment spiking neuron model for long-term sequential modelling." Proceedings of the AAAI conference on artificial intelligence. Vol. 38. No. 15. 2024.

**Questions:**

1. How does the method scale with spiking-based transformer networks (e.g., [1])?

2. Could the clipping operation limit the expressivity of the learned dynamics? Have the authors considered smooth alternatives (e.g., tanh)?

3. The paper mentions challenges in surrogate gradient tuning for LNMs. Is there a plan to extend recent adaptive surrogate methods to LNMs?

4. Could you provide a comparison of training time or convergence speed between LNM and LIF?

[1] "Spike-driven transformer." Advances in neural information processing systems 36 (2023): 64043-64058.

---

> ### Author Response · Authors · 2025-11-18
> **Response 1**
>
> Dear Reviewer XYhv,
>
> We sincerely appreciate your thoughtful feedback and the insights you provided on our work. Thank you for recognizing the practicality of our approach and for your positive remarks regarding our experimental suite, computational efficiency, and analysis of the learned dynamics across layers.
>
> Below are detailed comments regarding your concerns. We hope that these help clarify the contribution of our paper.
>
> # The paper could consider adding pseudocode to more clearly and intuitively demonstrate the differences between the proposed model and LIF.
>
> We agree that pseudocode can help us to better highlight the differences between our proposed method and LIF. Therefore, we have added pseudocode to Appendix B.
>
>
>
> # There is a lack of discussion regarding related work on neuron models [1-3].
>
> Thank you for bringing these missing related works to our attention. We have since updated Section 2.2 of our draft to include these works within our discussion.
>
>
> # How does the method scale with spiking-based transformer networks (e.g., [1])?
>
> Full-scale transformer experiments are not included in this rebuttal because we are not yet confident in our preliminary results and prefer not to present analyses that were rushed or insufficiently validated. Importantly, the ideology behind LNMs is architecture-agnostic, and our initial checks suggest that integrating it into transformer-based models requires minimal modification to their core architecture. We view a thorough investigation of applying LNM to transformer settings as a natural and valuable direction for future work. At this stage, we believe the presented results already demonstrate a meaningful and incremental contribution over prior research without relying on incomplete transformer experiments.
>
> # Could the clipping operation limit the expressivity of the learned dynamics? Have the authors considered smooth alternatives (e.g., tanh)?
>
> While it is true that clipping can constrain the expressiveness of the learned neuron dynamics, our original discussion in Section 6 (LIMITATIONS) focused primarily on this limitation from a theoretical perspective. There, we noted:
>
> "Our choice of low-degree polynomials may limit the expressiveness and variability of learned neuron models. Additionally, clipping the input to polynomials to the range $[-1, 1]$ can cause information loss, as values outside this range become indistinguishable. Therefore, finding alternatives to polynomials that don't significantly increase energy consumption or computational cost, or decrease model stability, is important for future research."
>
> However, we did not empirically examine how different clipping strategies influence the behavior of the learned dynamics. To address this gap, we conducted a small ablation study on CIFAR-100 using four timesteps and evaluated three bounding methods for LNM: (1) the standard [-1,1] clipping used throughout this work, (2) a much wider clipping window of [-5,5], and (3) the tanh function as a suggested alternative. The results of this comparison are summarized in the table below.
>
> | **Dataset** | **Bounding Method** | **Architecture** | **Timesteps** | **Accuracy** |
> |-------------|----------------------|-------------------|---------------|--------------|
> | CIFAR-100   | TanH                 | ResNet-19         | 4             | 80.52 ± 0.08% |
> | CIFAR-100   | Clip [-5, 5]         | ResNet-19         | 4             | 80.61 ± 0.09% |
> | CIFAR-100   | Clip [-1, 1]         | ResNet-19         | 4             | **80.70 ± 0.14%** |
>
> Given these results, we observe that the choice of clipping method has a non-trivial effect on model performance. Notably, expanding the clip range consistently reduces accuracy. This trend aligns with our discussion in Section 4.2.1 (Choice of Parameterization): low-degree polynomial neuron models have limited expressivity, and restricting the input domain helps keep the polynomial in a region where it can better approximate the desired non-linear behavior. When the input range is widened, the polynomial must represent a larger portion of the function space, leading to approximation error and degraded performance.
>
> For TanH, we hypothesize that its inherent smoothness suppresses sharp voltage transitions. While this stabilizes the dynamics, it may also dampen the neuron's ability to respond strongly to informative features. In contrast, clipping the activation to [-1,1] yields more abrupt voltage transitions within the membrane potential’s critical zone, which may help the neuron better distinguish salient inputs.
>
> We have added this discussion to Appendix C.

---

> > ### Author Response · Authors · 2025-11-20
> > **Question 2 Response 2**
> >
> > # Could the clipping operation limit the expressivity of the learned dynamics? Have the authors considered smooth alternatives (e.g., tanh)?
> >
> > Given your question, we also wanted to assess how different bounding strategies for LNM affect datasets where temporal credit assignment is crucial. We compare the same three bounding strategies for LNM: (1) the standard $[-1, 1]$ clipping used throughout this work, (2) a much wider $[-5, 5]$ clipping range, and (3) the tanh function as a smooth, non-linear alternative. However, this time we use CIFAR-10 DVS. The results are shown in the table below.
> >
> > | Dataset         | Bounding Method | Architecture | Timesteps | Accuracy              |
> > |-----------------|-----------------|--------------|-----------|------------------------|
> > | CIFAR-10 DVS   | TanH            | ResNet-19    | 10         | 80.90 ± 0.42%          |
> > | CIFAR-10 DVS    | Clip [-5, 5]    | ResNet-19    | 10         | 74.79 ± 1.98%          |
> > | CIFAR-10 DVS   | Clip [-1, 1]    | ResNet-19    | 10         | **81.39 ± 0.25%**      |
> >
> > Similarly to CIFAR-100 (T = 4), we see that Clip [-1, 1] outperforms the other bounding methods. However, this time, the larger Clip window of [-5, 5] appears to be more detrimental to learning. This is most likely due to the longer time neurons have to allow their voltage to evolve. Once again, TanH performs well and falls just short of Clip [-1, 1], which we believe to be for the same reasons discussed in our first response.

---

> ### Author Response · Authors · 2025-11-18
> **Response 2**
>
> ## Tables 1 and 2 should include results for the LIF model obtained under the same training conditions and architecture as LNM, to allow for a direct and intuitive comparison of the performance improvement.
>
> Thank you for pointing out this lack of results and comparison for the LIF neuron. To remedy this, we have performed several new experiments using the LIF neuron under the exact same experimental setup as the LNM neuron model. That is, the LIF neuron has the same decay, threshold, and surrogate gradient window as the LNM neuron model. The following table compares the results of both these neuron models, CIFAR-10, CIFAR-100, and ImageNet. Similarly, we used ResNet-19 for CIFAR-10/CIFAR-100 and ResNet-34 for ImageNet.
>
> | **Neuron Model**                  | **Timesteps**   | **CIFAR-10**              | **CIFAR-100**             | **ImageNet**              |
> |---------------------------|------------------|---------------------------|---------------------------|---------------------------|
> | LIF (Our Implementation)  | 4/4/4            | 96.20 ± 0.20%             | 78.95 ± 0.02%             | 69.04%                    |
> | LIF (Our Implementation)  | 2/2/×           | 96.16 ± 0.01%             | 79.92 ± 0.10%             | ×                         |
> | **LNM**            | **4/4/4**        | **97.01 ± 0.04%**         | **80.70 ± 0.14%**         | **70.91% ± 0.03%**        |
> | **LNM**            | **2/2/×**       | **96.96 ± 0.10%**         | **80.07 ± 0.13%**         | ×                         |
>
> On CIFAR-10, LNM outperforms LIF by 0.8% and 0.81% with two and four timesteps, respectively. On CIFAR-100, LNM achieves 0.15% higher accuracy with two timesteps and 1.75% higher accuracy with four timesteps, compared to LIF. Finally, on ImageNet, LNM outperforms LIF by a substantial 1.87%.
>
> Next, let's compare the two neuron models on CIFAR-10 DVS. Looking at the following table, we see that with VGGSNN, LNM outperforms LIF by 7.85%. LNM also outperforms LIF on ResNet-19 by 8.23%. While these differences initially seemed too large, they are in line with the reported values shown in [1], [2], and [3].
>
>
> | **Dataset**| **Neuron Model**                 | **Architecture** | **Timesteps** | **Accuracy**           |
> |-------------|---------------------------|-------------------|---------------|-------------------------|
> | CIFAR-10 DVS    | LIF (Our Implementation)  | VGGSNN            | 10            | 75.10 ± 1.31%          |
> | CIFAR-10 DVS   | LIF (Our Implementation)  | ResNet-19         | 10            | 73.16 ± 1.37%          |
> | CIFAR-10 DVS   | **LNM**                   | **VGGSNN**        | **10**        | **82.95 ± 0.69%**      |
> | CIFAR-10 DVS   | **LNM**            | **ResNet-19**     | **10**        | **81.39 ± 0.25%**      |
>
> We have added these new results to Tables 1 and 2 of our work, which we feel have strengthened the performance claims of LNMs over LIF.
>
> During this rebuttal period, we trained an LIF neuron under the same settings as our LNM and have updated the results in Table 1 and Table 2 to include such comparisons. We thank you for pointing out this weakness in our work and hope that this helps to better showcase the performance improvement of LNM over LIF.
>
> [1] Bhaskar Mukhoty, Velibor Bojkovic, William de Vazelhes, Xiaohan Zhao, Giulia De Masi, Huan Xiong, and
> Bin Gu. Direct training of SNN using local zeroth order method. In Thirty-seventh Conference on Neural
> Information Processing Systems, 2023.
>
> [2] Yufei Guo, Yuhan Zhang, Yuanpei Chen, Weihang Peng, Xiaode Liu, Liwen Zhang, Xuhui Huang, and Zhe
> Ma. Membrane potential batch normalization for spiking neural networks. In 2023 IEEE/CVF International
> Conference on Computer Vision (ICCV), pages 19363–19373, 2023.
>
> [3] Yufei Guo, Yuanpei Chen, Liwen Zhang, Xiaode Liu, Yinglei Wang, Xuhui Huang, and Zhe Ma. Im-loss:
> Information maximization loss for spiking neural networks. In S. Koyejo, S. Mohamed, A. Agarwal, D. Belgrave,
> K. Cho, and A. Oh, editors, Advances in Neural Information Processing Systems, volume 35, pages 156–166.
> Curran Associates, Inc., 2022.

---

> > ### Author Response · Authors · 2025-11-18
> > **Response 3**
> >
> > # The paper mentions challenges in surrogate gradient tuning for LNMs. Is there a plan to extend recent adaptive surrogate methods to LNMs?
> >
> > Thank you for raising this point. We actually explored this direction extensively early in the project. Our initial attempt was to generalize the approach of [4], which computes a surrogate window based on statistically derived decay-dependent moments of the membrane potential during training. While this extends naturally to our first- and second-degree LNMs, applying it to an $N$-degree LNM with $N>2$ requires computing moments up to order $N$. Even for $N=3$, the computational overhead becomes substantial, making this approach impractical for higher-degree models.
> >
> > Alternative adaptive surrogate-gradient methods, such as [5], could alleviate some of this cost. However, in our experiments, the surrogate gradient was not a dominant bottleneck, and therefore, we prioritized other aspects of the model. We agree that developing a robust technique for adaptive surrogate gradients is an interesting direction for future work.
> >
> > # Could you provide a comparison of training time or convergence speed between LNM and LIF?
> >
> > We have added the following discussion on training and inference time of LNM vs LIF to Section 5.4 (ENERGY EFFICIENCY AND RUNTIME) of our work. Here is the updated discussion:
> >
> > "Lastly, we discuss the training and inference overheads observed in our experimental setup for degree-3 LNMs. Across all experiments, training incurred an average runtime overhead of approximately 42\%, while inference exhibited a much smaller runtime overhead of around 4\%. These results indicate that the additional learnable parameters non-trivially impact training time. However, since inference constitutes the primary workload of a model during deployment, the relatively modest increase in inference time suggests that the added complexity of LNMs remains practical for real-world applications.
> >
> > It is worth noting that this inference overhead should be viewed as an upper bound, since deploying LNMs on event-driven hardware would require only active neurons to compute the update. However, in our GPU-based experimental setup, the update for every neuron, whether it receives input or not, must be computed. Approximating the overhead of event-driven hardware remains difficult due to limited availability."
> >
> > [4] Shuang Lian, Jiangrong Shen, Qianhui Liu, Ziming Wang, Rui Yan, and Huajin Tang. Learnable surrogate
> > gradient for direct training spiking neural networks. In Proceedings of the Thirty-Second International Joint
> > Conference on Artificial Intelligence, page 3002–3010, Macau, SAR China, August 2023. International Joint
> > Conferences on Artificial Intelligence Organization.
> >
> > [5] Shikuang Deng, Hao Lin, Yuhang Li, and Shi Gu. Surrogate module learning: reduce the gradient error
> > accumulation in training spiking neural networks. In Proceedings of the 40th International Conference on
> > Machine Learning, ICML’23. JMLR.org, 2023.

---

### Author Response · Authors · 2025-11-17
**Update on Comment Status**

Dear Reviewers,

Thank you for your thoughtful comments and questions. We have been working on addressing each point in detail. Please expect our initial responses in the coming days. As some of the questions require additional experiments, those answers may take slightly longer to finalize.

We appreciate your patience and look forward to continuing the discussion.

Thank you.

---

> ### Author Response · Authors · 2025-11-18
>
> Dear Reviewers,
>
> We have provided an initial set of responses to your questions and concerns. Some items remain unaddressed, as they depend on results from ongoing experiments. We will update our responses as these experiments are completed.
>
> We look forward to discussing these responses with you all.

---

### Meta-Review · Area_Chair_JV8j · 2025-12-19

**Summary:**

This paper proposes Learnable Neuron Models (LNMs), which parameterize spiking neuron dynamics with low-degree polynomials learned end-to-end. While the method shows consistent accuracy gains over LIF baselines across several datasets, reviewers converge on a critical concern: the paper lacks sufficient depth in analyzing why the proposed parameterization works and whether it represents a meaningful conceptual advance over simply tuning existing neuron model parameters. The performance improvements, though consistent, are modest on static datasets and come with increased complexity, without a clear demonstration of novel temporal modeling capability or a theoretical insight that justifies the added cost.

**Reviewer Concerns:**

Addressed:

Direct LIF Comparison (XYhv, TxX4, ZGV9): Authors added comprehensive LIF baseline experiments, showing clear accuracy gains.

Clarity & Presentation (XYhv, WbvZ): Pseudocode added, writing errors corrected.

Energy Overhead Calculation (WbvZ): Detailed computational breakdown provided.

Additional Experiments (ZGV9, WbvZ): Added results on psMNIST and ablation on clipping/bounding methods.

Outstanding (Core Reasons for Rejection):

Insufficient Analysis & Conceptual Depth (ZGV9, XYhv): The central weakness remains. The paper shows that LNMs work but lacks a deep analysis of why specific learned dynamics emerge and how they functionally improve network processing beyond acting as a better-tuned regularizer. The visualizations of diverse dynamics are descriptive, not explanatory.

Marginal/Uncertain Benefit on Temporal Tasks (ZGV9): The improvement on the key temporal benchmark (psMNIST) is very small (0.55%). A reviewer expressed skepticism about the unusually high LIF baseline, casting doubt on the significance of the gain. The paper fails to demonstrate that the learned polynomial dynamics confer a decisive advantage for modeling long-term dependencies, which is a key motivation for moving beyond LIF.

Limited Innovation & Hyperparameter Trade-off (WbvZ, XYhv): The core technique—using a learned polynomial—is seen by some reviewers as an incremental parameterization change rather than a fundamental innovation. The introduction of the polynomial degree as a new hyperparameter is acknowledged, and the benefits (especially at lower timesteps) appear modest relative to this added complexity.

Narrow Scope (ZGV9, XYhv): Evaluations are confined to convolutional architectures. The authors explicitly declined to test on spiking transformers, which limits claims about the general applicability of the method to modern SNN architectures.

**Reviewer Scores:**

Reviewer XYhv (Initial: 2, Reject): Concerns about innovation depth and analysis were not substantively resolved. Would maintain a score of 2.

Reviewer WbvZ (Initial: 4): Although pleased with clarifications, the reviewer's underlying score reflected reservations about innovation. The outstanding issues align with this view. Final score likely remains 4.

Reviewer TxX4 (Initial: 6 → 8): This reviewer's primary concern (LIF comparison) was fully addressed, leading them to raise their score. However, their positive view is an outlier against the consensus.

Reviewer ZGV9 (Initial: 4): Explicitly stated that weaknesses regarding analysis depth and temporal task evaluation were insufficiently addressed. Would maintain a score of 4.

---

### Decision · Program_Chairs · 2026-01-26

Reject